# The long non-coding RNA *Paupar* promotes KAP1-dependent chromatin changes and regulates olfactory bulb neurogenesis

Ioanna Pavlaki[1],[†] (ID), Farah Alammari[2],[†], Bin Sun[2], Neil Clark[3], Tamara Sirey[3], Sheena Lee[2], Dan J Woodcock[4], Chris P Ponting[3], Francis G Szele[2] & Keith W Vance[1],[*] (ID)

## Abstract

Many long non-coding RNAs (lncRNAs) are expressed during central nervous system (CNS) development, yet their *in vivo* roles and mechanisms of action remain poorly understood. *Paupar*, a CNS-expressed lncRNA, controls neuroblastoma cell growth by binding and modulating the activity of transcriptional regulatory elements in a genome-wide manner. We show here that the *Paupar* lncRNA directly binds KAP1, an essential epigenetic regulatory protein, and thereby regulates the expression of shared target genes important for proliferation and neuronal differentiation. *Paupar* promotes KAP1 chromatin occupancy and H3K9me3 deposition at a subset of distal targets, through the formation of a ribonucleoprotein complex containing *Paupar*, KAP1 and the PAX6 transcription factor. *Paupar*-KAP1 genome-wide co-occupancy reveals a fourfold enrichment of overlap between *Paupar* and KAP1 bound sequences, the majority of which also appear to associate with PAX6. Furthermore, both *Paupar* and *Kap1* loss-of-function *in vivo* disrupt olfactory bulb neurogenesis. These observations provide important conceptual insights into the *trans*-acting modes of lncRNA-mediated epigenetic regulation and the mechanisms of KAP1 genomic recruitment, and identify *Paupar* and *Kap1* as regulators of neurogenesis *in vivo*.

**Keywords** chromatin; gene regulation; KAP1; lncRNA; neurogenesis
**Subject Categories** Chromatin, Epigenetics, Genomics & Functional Genomics; Neuroscience; RNA Biology
**The EMBO Journal (2018) 37: e98219**

## Introduction

A subset of nuclear long non-coding RNAs (lncRNAs) have been shown to act as transcription and chromatin regulators using various mechanisms of action. These include local functions close to the sites of lncRNA synthesis (Engreitz *et al*, 2016) as well as distal modes of action across multiple chromosomes (Chalei *et al*, 2014; Vance *et al*, 2014). Moreover, lncRNA regulatory effects may be mediated by the act of lncRNA transcription as well as RNA sequence-dependent interactions with transcription factors and chromatin-modifying proteins (Vance & Ponting, 2014; Rutenberg-Schoenberg *et al*, 2016). Some lncRNAs have been proposed to act as molecular scaffolds to facilitate the formation of multicomponent ribonucleoprotein regulatory complexes (Tsai *et al*, 2010; Zhao *et al*, 2010; Ilik *et al*, 2013; Maenner *et al*, 2013; Yang *et al*, 2014b), whilst others may act to guide chromatin-modifying complexes to specific binding sites genome-wide (Vance & Ponting, 2014). Studies of *cis*-acting lncRNAs such as *Haunt* and *Hottip* have shown that lncRNA transcript accumulation at their sites of expression can effectively recruit regulatory complexes (Yin *et al*, 2015; Pradeepa *et al*, 2017). LncRNAs, however, have also been reported to directly bind and regulate genes across multiple chromosomes away from their sites of synthesis (Chu *et al*, 2011; Chalei *et al*, 2014; Vance *et al*, 2014; West *et al*, 2014; Carlson *et al*, 2015). By way of contrast, the mechanisms by which such *trans*-acting lncRNAs mediate transcription and chromatin regulation at distal bound target genes are less clear.

LncRNAs show a high propensity to be expressed in various brain regions and cell types relative to other tissues (Mercer *et al*, 2008, 2010; Ponjavic *et al*, 2009). The adult neurogenic stem cell-containing mouse subventricular zone (SVZ) generates neurons throughout life, contributes to brain repair and can be stimulated to limit damage, but is also a source of tumours (Bardella *et al*, 2016; Chang *et al*, 2016). During SVZ lineage progression, neural stem cells give rise to transit amplifying progenitors which in turn generate neuroblasts that migrate in the rostral migratory stream (RMS) to the olfactory bulbs (OB; Doetsch *et al*, 1999). The neuroblasts primarily become granule neurons that differentiate by extending long branched dendritic processes towards the glomerular layer (Petreanu & Alvarez-Buylla, 2002). There they integrate into and

1 Department of Biology and Biochemistry, University of Bath, Bath, UK
2 Department of Physiology, Anatomy and Genetics, University of Oxford, Oxford, UK
3 MRC Human Genetics Unit, The Institute of Genetics and Molecular Medicine, Western General Hospital, University of Edinburgh, Edinburgh, UK
4 Warwick Systems Biology Centre, University of Warwick, Coventry, UK
*Corresponding author. Tel: +44 1225 385106; E-mail: k.w.vance@bath.ac.uk
†These authors contributed equally to this work

modulate circuitry connecting peripheral olfactory receptor neurons with the output neurons of the OB (Gheusi *et al*, 2000; Lledo & Saghatelyan, 2005). It has been estimated that 8,992 lncRNAs are expressed in the SVZ neurogenic system, many of which are differentially expressed during SVZ/OB neurogenesis, suggesting that at least some of these transcripts may play regulatory roles (Ramos *et al*, 2013). However, only a minority of SVZ expressed lncRNAs have been analysed functionally and the full scope of their molecular mechanisms of action remain poorly understood.

*Kap1* encodes an essential chromatin regulatory protein that plays a critical role in embryonic development and in adult tissues. Kap1$^{-/-}$ mice die prior to gastrulation while hypomorphic *Kap1* mouse mutants display multiple abnormal embryonic phenotypes, including defects in the development of the nervous system (Cammas *et al*, 2000; Herzog *et al*, 2011; Shibata *et al*, 2011). KAP1 interacts with chromatin binding proteins such as HP1 and the SETDB1 histone-lysine N-methyltransferase to control heterochromatin formation and to silence gene expression at euchromatic loci (Iyengar & Farnham, 2011). Despite this fundamental role in epigenetic regulation, the mechanisms of KAP1 genomic targeting are not fully understood. KAP1 does not contain a DNA binding domain but was originally identified through its interaction with members of the KRAB zinc finger (KRAB-ZNF) transcription factor family. Subsequent studies, however, revealed that KRAB–ZNF interactions cannot account for all KAP1 genomic recruitment events. KAP1 preferentially localises to the 3′ end of zinc finger genes as well as to many promoters and intergenic regions in human neuronal precursor cells. A mutant KAP1 protein, however, that is unable to interact with KRAB-ZNFs still binds to promoters, suggesting functionally distinct subdomains (Iyengar *et al*, 2011). This work points to the presence of alternative, KRAB-ZNF-independent, mechanisms that operate to target KAP1 to a distinct set of genomic binding sites. We reasoned that this may involve specific RNA–protein interactions between KAP1 and chromatin-bound lncRNAs.

The CNS-expressed intergenic lncRNA *Paupar* represents an ideal candidate chromatin-enriched lncRNA with which to further define *trans*-acting mechanisms of lncRNA-mediated gene and chromatin regulation. *Paupar* is transcribed upstream from the *Pax6* transcription factor gene and acts to control proliferation and differentiation of N2A neuroblastoma cells *in vitro* (Vance *et al*, 2014). *Paupar* regulates *Pax6* expression locally, physically associates with PAX6 protein and interacts with distal transcriptional regulatory elements to control gene expression on multiple chromosomes in N2A cells in a dose-dependent manner. Here, we show that *Paupar* directly interacts with KAP1 in N2A cells and that together they control the expression of a shared set of target genes enriched for regulators of neural proliferation and differentiation. Our findings indicate that *Paupar*, KAP1 and PAX6 physically associate on chromatin within the regulatory region of shared target genes and that *Paupar* knockdown reduces both KAP1 chromatin association and histone H3 lysine 9 trimethylation (H3K9me3) at PAX6 co-bound locations. Genome-wide occupancy maps further identified a fourfold enrichment in the overlap between *Paupar* and KAP1 binding sites on chromatin, the majority of which (73%) are also estimated to be bound by PAX6. Our results also show that both *Paupar* and KAP1 loss-of-function *in vivo* disrupt SVZ/OB neurogenesis. We propose that *Paupar* and *Kap1* are novel regulators of neurogenesis *in vivo* and that *Paupar* operates as a transcriptional cofactor to promote

KAP1-dependent chromatin changes at a subset of bound regulatory elements in *trans* via association with non-KRAB-ZNF transcription factors such as PAX6.

# Results

### *Paupar* directly binds the KAP1 chromatin regulatory protein in mouse neural cells in culture

The lncRNA *Paupar* binds transcriptional regulatory elements across multiple chromosomes to control the expression of distal target genes in N2A neuroblastoma cells (Vance *et al*, 2014). Association with transcription factors such as PAX6 assists in targeting *Paupar* to chromatin sites across the genome. As *Paupar* depletion does not alter PAX6 chromatin occupancy (Vance *et al*, 2014), we hypothesised that *Paupar* may recruit transcriptional cofactors to PAX6 and other neural transcription factors to regulate gene expression. To test this, we sought to identify transcription and chromatin regulatory proteins that bind both *Paupar* and PAX6 in N2A cells in culture. *In vitro*-transcribed biotinylated *Paupar* was therefore immobilised on streptavidin beads and incubated with N2A cell nuclear extract in a pulldown assay. Bound proteins were washed, eluted and identified using mass spectrometry (Fig 1A). This identified a set of 78 new candidate *Paupar*-associated proteins that do not bind a control RNA of similar size, including 28 proteins with annotated functions in the control of gene expression that might function as transcriptional cofactors (Fig 1B and Dataset EV1).

We next performed native RNA-IP experiments in N2A cells to validate potential associations between the endogenous *Paupar* transcript and five gene expression regulators. These candidates were as follows: RCOR3, a member of the CoREST family of proteins that interact with the REST transcription factor; KAP1, a key epigenetic regulator of gene expression and chromatin structure; PPAN, a previously identified regulator of *Pax6* expression in the developing eye; CHE-1, a polymerase II interacting protein that functions to promote cellular proliferation and block apoptosis; and ERH, a transcriptional cofactor that is highly expressed in the eye, brain and spinal cord.

The results revealed that the *Paupar* transcript, but not a non-specific control RNA, was > twofold enriched using antibodies against RCOR3, KAP1, ERH, PPAN or CHE1 compared to an IgG isotype control in a native RNA-IP experiment (Fig 1C). In addition, *Paupar* did not associate above background with SUZ12, EED and EZH2 Polycomb proteins used as negative controls. This served to further confirm the specificity of the *Paupar* lncRNA–protein interactions because Polycomb proteins associate with a large number of RNAs (Davidovich *et al*, 2015) and yet were not identified as *Paupar* interacting proteins in our pulldown assay. The endogenous *Paupar* transcript therefore associates with proteins involved in transcription and chromatin regulation in proliferating N2A cells.

To characterise *Paupar* lncRNA–protein interactions further, we used UV-RNA-IP to test whether *Paupar* interacts directly with any of these five cofactors. These data showed that *Paupar*, but not an *U1snRNA* control, is highly enriched using antibodies against KAP1 or RCOR3 compared to an IgG control (Fig 1D). A lower level of *Paupar* enrichment is found with CHE1, whereas ERH or PPAN does not appear to interact directly with *Paupar* (Fig EV1). Furthermore,

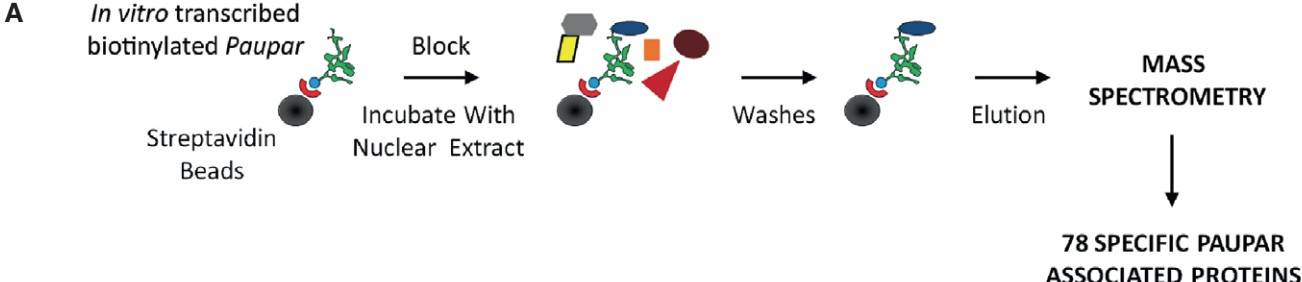

| GOID | Term | Number In Set | Number In Reference | Corrected P-value |
|---|---|---|---|---|
| GO:0006396 | RNA processing | 19 | 513 | $3.35 \times 10^{-12}$ |
| GO:0008380 | RNA splicing | 13 | 237 | $6.72 \times 10^{-10}$ |
| GO:0022613 | ribonucleoprotein complex biogenesis | 10 | 195 | $5.87 \times 10^{-7}$ |
| GO:0010467 | gene expression | 28 | 3596 | $5.44 \times 10^{-3}$ |

Figure 1.

the association of *Paupar* with either KAP1 or RCOR3 was reduced in the absence of UV treatment (Fig 1E). These results indicate that the endogenous *Paupar* transcript directly and specifically associates with RCOR3 and KAP1 transcriptional cofactors in neural precursor-like cells in culture.

As a first step to determine whether KAP1 or RCOR3 can act as PAX6-associated transcriptional cofactors, we performed immuno-precipitation experiments in N2A cells using transfected FLAG-tagged PAX6 and KAP1 or RCOR3 proteins. Immunoprecipitation of FLAG-PAX6 using anti-FLAG beads co-immuno-precipitated transfected KAP1 protein, but not RCOR3 (Fig 1F), suggesting that PAX6 and KAP1 are present within the same multicomponent regulatory complex. Consistent with this, a previous study showed that KAP1 interacts with PAX3 through the amino terminal paired domain, which is structurally similar in PAX6, to mediate PAX3-dependent transcriptional repression (Hsieh *et al*, 2006). Together, these results indicate that KAP1 may regulate *Paupar* and PAX6-mediated gene expression programmes.

## *Paupar* and KAP1 control expression of a shared set of target genes that are enriched for regulators of neuronal function and cell cycle in N2A cells

KAP1 regulates the expression of genes involved in the self-renewal and differentiation of multiple cell types, including neuronal cells (Iyengar & Farnham, 2011), and thus is an excellent candidate interactor for mediating the transcriptional regulatory function of *Paupar*. To investigate whether *Paupar* and KAP1 functionally interact to control gene expression, we first tested whether they regulate a common set of target genes. We depleted *Kap1* expression in N2A cells using shRNA transfection and achieved ~90% reduction in both protein (Fig 2A) and transcript (Fig 2B) levels. *Paupar* levels do not change upon KAP1 knockdown, indicating that KAP1-dependent changes in gene expression are not due to regulation of *Paupar* expression (Fig 2B). Transcriptome profiling using microarrays then identified 1,913 differentially expressed genes whose expression significantly changed [at a 5% false discovery rate (FDR)] greater than 1.4-fold (log2 fold change ≈ 0.5) upon KAP1 depletion (Fig 2C

and Dataset EV2). 282 of these genes were previously identified to be regulated by human KAP1 in Ntera2 undifferentiated human neural progenitor cells (Iyengar *et al*, 2011). Transient reduction in *Kap1* expression by ~55% using a second shRNA expression vector (*Kap1* shB) also induced expression changes for seven out of eight KAP1 target genes with known functions in neuronal cells that were identified in the microarray (Fig EV2). These data further validate the specificity of the KAP1 regulated gene set.

We previously showed that *Paupar* knockdown induces changes in the expression of 942 genes in N2A cells (Vance *et al*, 2014). Examination of the intersection of KAP1 and *Paupar* transcriptional targets identified 244 genes whose levels are affected by both *Paupar* and KAP1 knockdown in this cell type (Fig 2D and Dataset EV3). This represents a significant 3.6-fold enrichment over the number expected by random sampling and is not due to co-regulation because *Kap1* is not a *Paupar* target (Vance *et al*, 2014). A large majority (87%; 212/244) of these common targets are positively regulated by *Paupar* and for two-thirds of these genes (161/244) their expression changes in the same direction upon *Paupar* or KAP1 knockdown (Fig 2E). Furthermore, Gene Ontology enrichment analysis of these 244 genes showed that *Paupar* and KAP1 both regulate a shared set of target genes enriched for regulators of interphase, components of receptor tyrosine kinase signalling pathways as well as genes involved in nervous system development and essential neuronal cell functions such as synaptic transmission (Fig 2F). Genes targeted by both *Paupar* and KAP1 are thus expected to contribute to the control of neural stem cell self-renewal and neural differentiation.

## *Paupar*, KAP1 and PAX6 associate on chromatin within the regulatory region of shared target genes

In order to investigate *Paupar*-mediated mechanisms of distal gene regulation, we next sought to determine whether *Paupar*, KAP1 and PAX6 can form a ternary complex on chromatin within the regulatory regions of their shared target genes. To do this, we first integrated our analysis of PAX6-regulated gene expression programmes in N2A cells (Vance *et al*, 2014) and identified 87 of the 244 *Paupar* and KAP1 common targets, which is 35.8-fold

◀

**Figure 1. *Paupar* directly binds the KAP1 chromatin regulatory protein in mouse N2A neuroblastoma cells.**

A    Overview of the pulldown assay. *In vitro*-transcribed biotinylated *Paupar* RNA was immobilised on streptavidin beads and incubated with N2A cell nuclear extract. Bound RNA protein complexes were extensively washed and specific *Paupar*-associated proteins, which do not interact with a control mRNA of a similar size, identified by mass spectrometry.

B    Gene Ontology terms were used to annotate *Paupar*-associated proteins according to biological process. The Bonferroni correction was used to adjust the *P*-values to correct for multiple testing.

C    Endogenous *Paupar* transcript interacts with transcription and chromatin regulatory proteins in N2A cells. *Paupar* association with the indicated proteins was measured using native RNA-IP. Whole cell lysates were prepared and the indicated regulatory proteins immuno-precipitated using specific antibodies. Bound RNAs were purified and the levels of *Paupar* and the *U1snRNA* control detected in each RIP using qRT–PCR. *Paupar* transcript directly interacts with KAP1 and RCOR3 in N2A cells.

D, E    Nuclear extracts were prepared from UV cross-linked (D) and untreated (E) cells and immuno-precipitated using either anti-KAP1, anti-RCOR3 or a rabbit IgG control antibody. Associated RNAs were stringently washed and purified. The levels of *Paupar* and the *U1snRNA* control transcript were detected in each UV-RIP using qRT–PCR.

F    PAX6 associates with KAP1 in N2A cells. FLAG-PAX6 and KAP1 or RCOR3 expression vectors were transfected into N2A cells. Lysates were prepared 2 days after transfection and FLAG-PAX6 protein immuno-precipitated using anti-FLAG beads. Co-precipitated proteins were detected by Western blotting.

Data information: For RNA-IP and UV-RIP assays, results are presented as fold enrichment relative to control antibody. Mean values ± SEM, *N* = 3. One-tailed *t*-test, unequal variance *P < 0.05, **P < 0.01, ***P < 0.001.

Source data are available online for this figure.

**Figure 2.** *Paupar* and KAP1 regulate shared target genes involved in neural cell proliferation and function.

N2A cells were transfected with either the shA *Kap1* targeting shRNA expression vector or a scrambled control and pTK-Hyg selection plasmid. Three days later, cells were expanded and hygromycin was added to the medium to remove untransfected cells.

A   After 7 days, Western blotting was performed to determine KAP1 protein levels. Lamin B1 was used as a loading control.
B   *Kap1* and *Paupar* transcript levels were analysed by qRT–PCR. Data were normalised using *Gapdh,* and expression changes are shown relative to a non-targeting scrambled control (set at 1). Mean values ± SEM, N = 3. One-tailed *t*-test, unequal variance **P < 0.01.
C   KAP1 regulated genes were identified using a GeneChip Mouse Gene 1.0 ST Array (5% FDR, log2 fold change > 0.5).
D   Intersection of *Kap1*- and *Paupar*-regulated genes revealed common target genes whose expression is controlled by both these factors.
E   The majority (87%) of *Paupar* and *Kap1* shared target genes are positively regulated by *Paupar*.
F   Gene Ontology analysis of *Paupar* and *Kap1* common target genes was performed using GOToolBox. Representative significantly enriched categories were selected from a hypergeometric test with a Benjamini–Hochberg-corrected *P*-value threshold of 0.05.

greater than expected by random sampling, whose expression is also controlled by PAX6 (Fig 3A and Dataset EV3). We found that 34 of these genes contain a CHART-Seq mapped *Paupar* binding site within their GREAT defined putative regulatory regions (Vance *et al*, 2014; Vance, 2016) and predicted that these represent functional *Paupar* binding events within close genomic proximity to direct transcriptional target genes (Fig 3A and Dataset EV3).

ChIP-qPCR analysis previously identified four of these *Paupar* bound locations within the regulatory regions of the *Mab21L2*, *Mst1*, *E2f2* and *Igfbp5* genes that are also bound by PAX6 in N2A cells (Vance *et al*, 2014). We therefore measured KAP1 chromatin occupancy at these regions as well as at a negative control sequence within the first intron of *E2f2* using ChIP and identified a specific enrichment of KAP1 chromatin association at the *Mab21L2*, *Mst1*, *E2f2* and *Igfbp5* genes compared to an IgG isotype control (Fig 3B). KAP1 binding to these regions is only

two- to fourfold reduced compared to the Zfp382 3′ UTR-positive control (Fig 3B), which represents an exemplar high-affinity KAP1 binding site (Iyengar *et al*, 2011). KAP1 and *Paupar* also co-occupy a binding site within the *Ezh2* gene. *Ezh2* is regulated by *Paupar* and KAP1 but not by PAX6 suggesting that transcription factors in addition to PAX6 may also be involved in modulating *Paupar*-KAP1 function. However, taken together these data indicate that *Mab21L2*, *Mst1*, *E2f2* and *Igfbp5* are co-ordinately regulated by a ribonucleoprotein complex containing *Paupar*-KAP1–PAX6.

### *Paupar* functions as a transcriptional cofactor to promote KAP1 chromatin occupancy and H3K9me3 deposition at PAX6 bound sequences

KAP1 is recruited to its target sites within 3′ UTRs of ZNF genes through association with KRAB-ZNF transcription factors (O'Geen

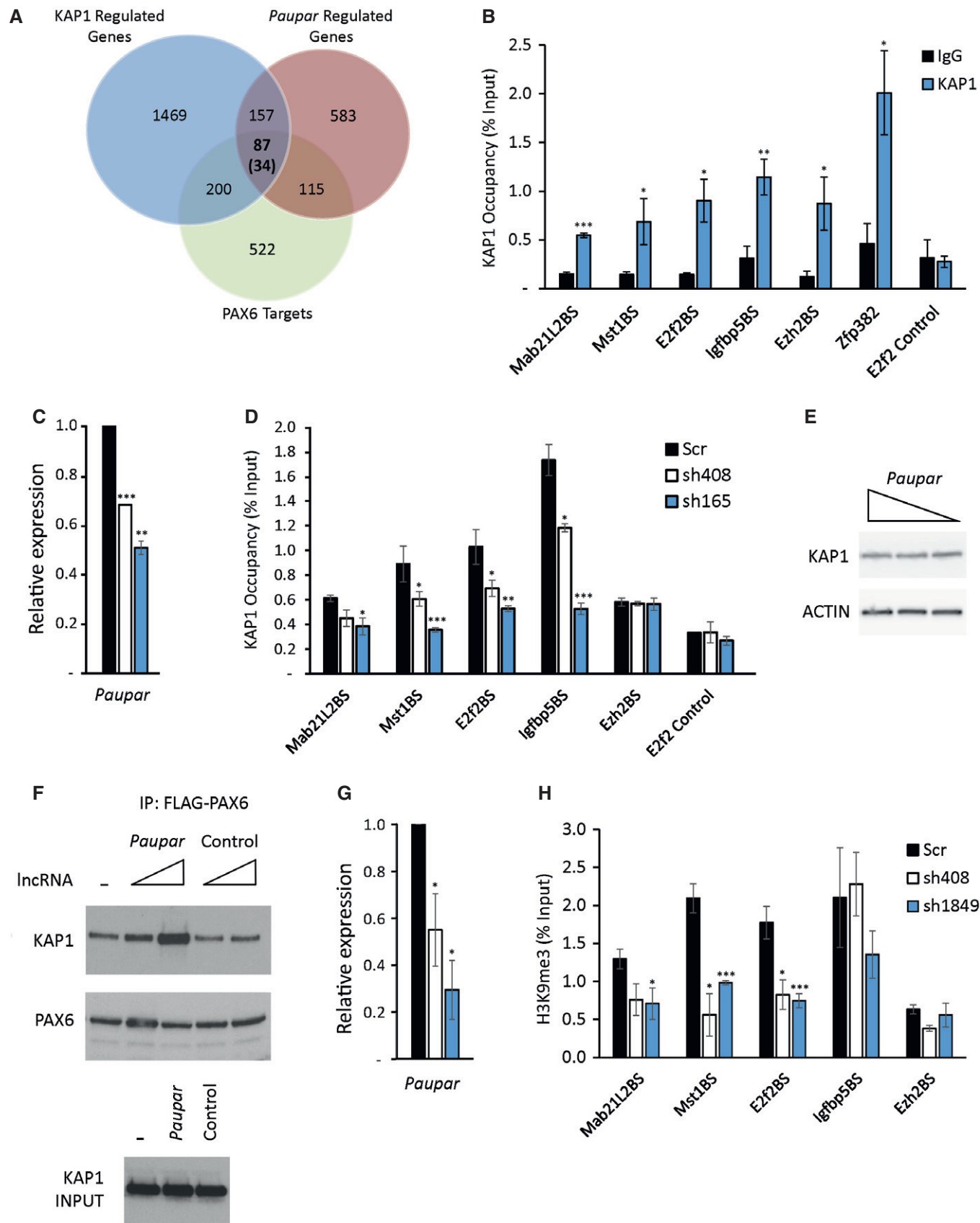

**Figure 3.**

◀

**Figure 3.  *Paupar* promotes KAP1 chromatin occupancy and H3K9me3 deposition at PAX6 bound sequences within the regulatory regions of common targets.**

A    Intersection of *Paupar*, KAP1 and PAX6 regulated genes identified 87 common target genes. 34 of these genes (in brackets) contain a *Paupar* binding site within their regulatory regions.

B    ChIP assays were performed in N2A cells using either an antibody against KAP1 or an isotype-specific control.

C    N2A cells were transfected with either a non-targeting control or two independent *Paupar* targeting shRNA expression vectors. Cells were harvested for ChIP 3 days later, and *Paupar* depletion was confirmed using qRT–PCR.

D    *Paupar* knockdown reduces KAP1 chromatin occupancy at shared binding sites. ChIP assays were performed 3 days after shRNA transfection using an anti-KAP1 polyclonal antibody.

E    Western blotting showed that KAP1 proteins levels do not change upon *Paupar* knockdown. Actin was used as a control.

F    *Paupar* promotes KAP1–PAX6 association. FLAG-PAX6 and KAP1 expression vectors were co-transfected into N2A cells along with increasing concentrations of *Paupar* or a size-matched control lncRNA expression vector. Expression of the maximum concentration of either *Paupar* or control RNA in each IP does not alter KAP1 input protein levels (lower panel). Lysates were prepared 2 days after transfection and FLAG-PAX6 protein immuno-precipitated using anti-FLAG beads. The amount of DNA transfected was made equal in each IP using empty vector and proteins in each complex were detected by Western blotting.

G, H    *Paupar* knockdown reduces H3K9me3 at a subset of bound sequences in *trans*. ChIP assays were performed using an anti-H3K9me3 polyclonal antibody 3 days after transfection of the indicated shRNA expression vectors.

Data information: For ChIP assays, the indicated DNA fragments were amplified using qPCR. % input was calculated as $100 \times 2^{(C_t \text{Input} - C_t \text{IP})}$. Results are presented as mean values ± SEM, $N = 3$. One-tailed *t*-test, unequal variance *$P < 0.05$, **$P < 0.01$, ***$P < 0.001$.

Source data are available online for this figure.

---

*et al*, 2007; Iyengar *et al*, 2011). However, *Paupar* bound sequences are preferentially located at gene promoters and are not enriched for KRAB-ZNF transcription factor binding motifs as determined using *de novo* motif discovery (Vance *et al*, 2014). This suggests that *Paupar* may play a role in recruiting KAP1 to a separate class of binding site in a KRAB-ZNF-independent manner. To test this, *Paupar* expression was first depleted using transient transfection of *Paupar* targeting shRNA expression vectors (Fig 3C). ChIP-qPCR was then performed to measure KAP1 chromatin occupancy in control and *Paupar* knockdown N2A cells at the four *Paupar*-KAP1–PAX6 co-occupied binding sites within the regulatory regions of the *Mab21L2, Mst1, E2f2* and *Igfbp5* genes, a *Paupar*-KAP1 bound sequence within the *Ezh2* gene that is not regulated by PAX6, and a control sequence that is not bound by *Paupar*. The results show that KAP1 chromatin binding is significantly decreased at the four *Paupar*-KAP1–PAX6 bound regions upon *Paupar* depletion and that the extent of KAP1 chromatin association appears to be dependent on *Paupar* transcript levels (Fig 3D). KAP1 chromatin association is also not reduced at the *Ezh2* gene *Paupar*-KAP1 binding site or at the control sequence that is not bound by *Paupar* (Fig 3D), whilst total KAP1 protein levels do not detectably change upon *Paupar* knockdown (Fig 3E), further confirming specificity.

These results imply that *Paupar* functions to promote KAP1 chromatin association at a subset of its genomic binding sites in *trans* and that this requires the formation of a DNA bound ternary complex containing *Paupar*, KAP1 and PAX6. Consistent with this, co-expression of *Paupar* promotes KAP1–PAX6 association in a dose-dependent manner in an immunoprecipitation experiment (Fig 3F). This effect is specific for the *Paupar* transcript because expression of a size-matched control RNA does not alter KAP1–PAX6 association. *Paupar* thus functions as a transcriptional cofactor to promote the assembly of a complex containing *Paupar*-KAP1–PAX6 on chromatin in *trans*. This ribonucleoprotein complex appears to function as a regulator of genes involved in controlling neural stem cell self-renewal and differentiation.

We next tested whether *Paupar* can induce histone modification changes at bound target genes on different chromosomes away from its sites of synthesis. As KAP1 interacts with the SETDB1 methyltransferase to mediate histone H3K9me3 deposition (Schultz *et al*, 2002), we first determined the levels of H3K9me3 at the shared binding sites near the *Mab21L2, Mst1, E2f2, Igfbp5* and *Ezh2* genes using ChIP-qPCR. This revealed an enrichment of H3K9me3-modified chromatin at all five locations (Fig EV3), consistent with a previous study showing that many KAP1 bound promoters are marked by H3K9me3 (O'Geen *et al*, 2007). ChIP analysis following *Paupar* depletion using two different shRNAs identified a significant decrease in histone H3K9me3 at three (*Mab21L2, Mst1* and *E2f2*) out of four shared binding sites within genes that are co-regulated by *Paupar*, KAP1 and PAX6 (Fig 3G and H). No change in histone H3K9me3 was detected at the *Ezh2* gene whose expression does not change upon PAX6 depletion. Together, these data show that *Paupar* functions to modulate KAP1 chromatin association and subsequent histone H3K9me3 deposition at a subset of its shared binding sites in *trans*.

Surprisingly, analysis of our microarray dataset of *Paupar*-mediated gene expression changes (Vance *et al*, 2014) showed that H3K9me3 deposition did not correlate with transcriptional repression. *Mab21L2* and *Mst1* were downregulated upon *Paupar* depletion suggesting that they are activated by the *Paupar*-KAP1–PAX6 complex whilst *E2f2* expression was upregulated, suggesting that it is repressed by this complex. These findings are consistent with recent work using dCas9 fusion proteins to target histone methylation to specific loci (O'Geen *et al*, 2017) and suggest a complex relationship between *Paupar*-mediated KAP1-dependent chromatin changes and gene expression.

### *Paupar* co-occupies an enriched subset of KAP1 binding sites genome-wide

We next examined the intersection between *Paupar* and KAP1 bound locations genome-wide in order to generate a more comprehensive view of the potential of *Paupar* for regulating KAP1 function. ChIP-seq profiling of KAP1 chromatin occupancy showed that KAP1 associates with 5,510 genomic locations compared to input DNA in N2A cells (1% FDR; Dataset EV4). KAP1 binding sites are particularly enriched at promoter regions, over gene bodies and at the 3′ UTRs of zinc finger genes (Fig 4A), consistent with previous studies mapping human KAP1 genomic occupancy (O'Geen *et al*, 2007; Iyengar *et al*, 2011). Intersection of KAP1 bound locations

with our CHART-seq map of *Paupar* genomic binding in N2A cells (Vance *et al*, 2014) identified 46 KAP1 binding sites that are co-occupied by *Paupar* and not bound in a LacZ-negative control CHART-seq pulldown (Fig 4B). Notably, this represents a significant ($P < 0.001$) fourfold enrichment of *Paupar* and KAP1 co-occupied locations as estimated using Genome Association Tester (GAT; Fig 4B). Plotting the distribution of peak intensities across these co-occupied regions revealed a precise coincidence of *Paupar* and KAP1 binding (Fig 4C). These data therefore show that *Paupar* co-occupies a specific subset of KAP1 bound sequences genome-wide.

We then examined the intersection between these 46 *Paupar*-KAP1 co-bound locations and the 244 *Paupar*-KAP1 co-regulated genes (Dataset EV3) and found shared binding sites within the putative regulatory regions of the *Npy*, *Syt1*, *Fam92b* and *Plxna4* genes. However, we expect this to be an under-representation of the total number of direct *Paupar*-KAP1 co-regulated targets given the complex cause-and-effect relationship between histone H3 methylation and gene expression (O'Geen *et al*, 2017).

Our analysis also revealed that only one of the 46 *Paupar*-KAP1 co-occupied sequences is located within the 3′ UTR of a ZNF gene (zfp68; Dataset EV4), pointing to an alternative mechanism of KAP1 genomic recruitment in addition to the well-described KRAB-ZNF association. To investigate this further, we performed ChIP-qPCR to interrogate the overlap between PAX6 and *Paupar*-KAP1 co-occupied locations. PAX6 occupancy was measured at a subset of ChIP-seq and CHART-seq defined KAP1-*Paupar* bound sequences as well as at previously identified *Paupar*-KAP1–PAX6 binding sites close to the *Mab21L2* and *Mst1* genes as positive controls. The results identified statistically significant PAX6 enrichment at 11 out of 15 (73%) locations tested compared to an IgG control (Fig 4D). These include PAX6 binding sites within the regulatory regions of four neuronal genes (*Npy*, *Syt1*, *Tshz2* and *Syt7*) whose expression changes when PAX6 expression is depleted in N2A cells (Vance *et al*, 2014). Taken together, these results indicate that PAX6 is likely to play a regulatory role at a large proportion of *Paupar*-KAP1 co-occupied sequences genome-wide. The absence of PAX6 from some of the tested locations further suggests that *Paupar*-KAP1 can associate with other transcription factors in addition to PAX6.

### *Paupar* and Kap1 regulate olfactory bulb neurogenesis

Our results indicate that *Paupar* and KAP1 regulate the expression of shared target genes important for proliferation and neuronal differentiation in N2A cells. We next expanded this observation and

tested whether *Paupar* and *Kap1* can regulate mouse SVZ/OB neurogenesis *in vivo*. In this system, neurogenesis can be monitored by electroporating neonatal SVZ neural stem cells and analysing differentiated neurons that have migrated into the OB 7 days post electroporation (dpe; Boutin *et al*, 2008; Chesler *et al*, 2008). RT–qPCR first showed that *Paupar* and *Kap1* are expressed in the SVZ and in neurospheres cultured from postnatal day 4 (P4) SVZ (Fig EV4A and B), consistent with *Kap1* expression data in the Allen Brain Atlas. Similar to N2A cells, shRNA expression vectors depleted *Paupar* and *Kap1* transcript in P4 SVZ neurospheres (Fig EV4C and D). Nucleofection of shRNA expression vectors targeted ~60% of cells, as measured using GFP, but we determined transcript levels in all cells. Thus, on a cell-by-cell basis the relative level of knockdown of both *Paupar* and *Kap1* is predicted to be greater than shown.

We then electroporated P1 pups with *Paupar* and *Kap1* shRNA expression constructs or a scrambled control and examined the number and morphology of neurons that migrated into the OB 7 dpe. The results showed that there were significantly fewer GFP$^+$ cells in the OB after *Paupar* knockdown (KD) using sh165 KD compared to control whilst KD with sh408 caused a slight but statistically non-significant decrease in OB GFP$^+$ cell numbers (Fig 5A and B). As sh165 more efficiently depletes *Paupar* expression compared to sh408 in N2A cells (Fig 3C) and in neurospheres (Fig EV4C), this result is suggestive of dose-dependent effects mediated by the *Paupar* transcript. Co-staining with the immature neuroblast marker DCX (Yang *et al*, 2004) showed that all GFP$^+$ cells in the OB were DCX$^+$, and this was not altered by *Paupar* KD (Fig EV4E). Similar to *Paupar*, at 7 dpe of either *Kap1* shRNA expression construct, there was a significant reduction in the number of GFP$^+$ cells that had migrated from the SVZ into the OB (Fig 5C and D). We controlled for apoptosis as this may lead to reduced cell numbers and did not detect changes in the percentage of GFP$^+$ cells that are TUNEL$^+$ at 3 or 7 dpe between scrambled control and any of the *Paupar* or *Kap1* shRNA expression vectors in the SVZ, RMS or OB (Fig EV5). These results therefore suggest that both *Paupar* and *Kap1* are required for the production of newborn OB neurons.

Interestingly, *Paupar* as well as *Kap1* knockdown altered the morphology of newborn neurons in the OB (Fig 5E–H). As expected (Petreanu & Alvarez-Buylla, 2002), in scrambled controls many GFP$^+$ neurons in the OB granule layer had begun morphological differentiation with processes extending radially towards the pial surface, some of which were branched. These cells were classified as class I (Fig 5E and F; Boutin *et al*, 2010). By contrast, after *Paupar* KD, a variety of abnormal morphologies

**Figure 4.  *Paupar* co-occupies a subset of KAP1 binding sites on chromatin genome-wide.**

5,510 KAP1 binding sites common to both replicates were identified relative to input DNA (1% FDR; Dataset EV4).

A   Sites of KAP1 occupancy are particularly enriched at promoter regions (5′ UTRs), over gene bodies and over the 3′ UTR exons of zinc finger genes [$q = 2 \times 10^{-5}$; GAT randomisation test (Heger *et al*, 2013)].

B   Intersection of KAP1 and *Paupar* binding sites in N2A cells identified 46 KAP1 bound locations that are specifically co-occupied by *Paupar*. This represents a significant fourfold enrichment [$P < 1 \times 10^{-3}$; GAT randomisation test (Heger *et al*, 2013)].

C   Sequencing read density distribution over the 46 shared binding locations was calculated and revealed a coincidence of *Paupar* and KAP1 binding site centrality. Colour legend indicates the base 2 logarithm of the ratio of read counts against input in bins of width 10 *nt*.

D   ChIP in N2A cells was performed using either an antibody against PAX6 or an isotype-specific control. *Paupar*-KAP1 co-occupied binding sites close to the indicated genes were amplified using qPCR to check for PAX6 chromatin association. % input was calculated as $100 \times 2^{(C_t\text{Input}-C_t\text{IP})}$. Results are presented as mean values $\pm$ SEM, $N = 3$. One-tailed *t*-test, unequal variance *$P < 0.05$, **$P < 0.01$, ***$P < 0.001$.

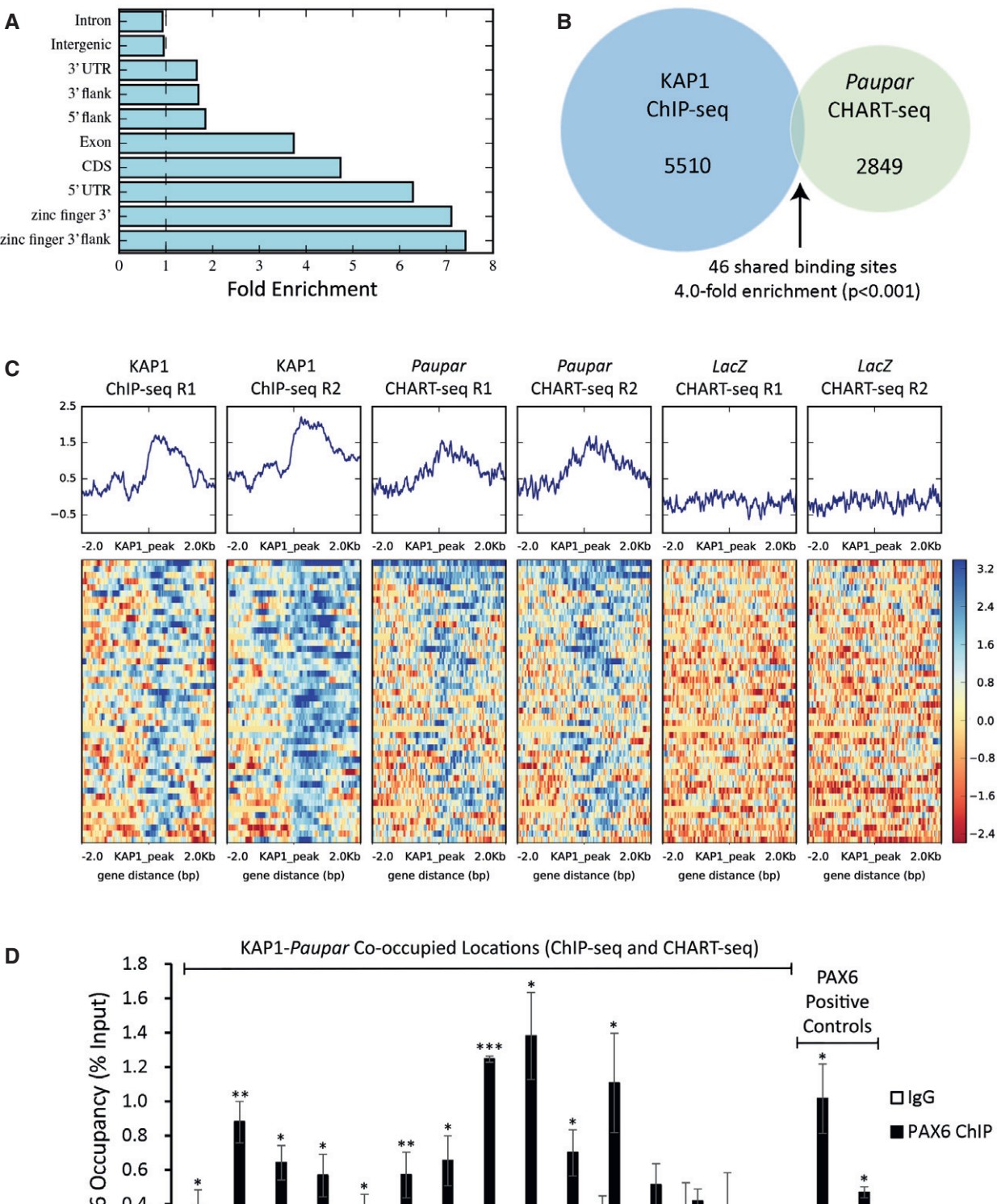

**Figure 4.**

were observed, which we classified as class II or class III (Fig 5E). Class II cells were rare but were distinguished by many short branched processes. Class III cells exhibited only short or no processes, suggesting they were still migrating or had not substantially differentiated (Fig 5E). Quantification revealed that after electroporation the percentage of cells with Class I morphology was $34 \pm 2\%$ in scrambled controls but only $8 \pm 3\%$ after *Paupar* KD with sh165 and $6 \pm 3\%$ in the sh408 group ($P = 0.0005$ and $P = 0.0009$, respectively; Fig 5G). Conversely, after *Paupar* KD there were significantly more class III neurons in the sh165 group ($87 \pm 4\%$) as well as in the sh408 group ($85 \pm 6\%$) compared to controls ($58 \pm 5\%$; $P = 0.003$ and $P = 0.02$, respectively). *Kap1* KD showed similar effects (Fig 5F and H); shA and shB resulted in $16.7 \pm 5.6$ and $19.3 \pm 2.0\%$ of Class I neurons versus $42.0 \pm 1.5\%$ in controls ($P = 0.012$ and $P = 0.013$, respectively). Again, the number of Class III neurons increased from $54.7 \pm 2.2\%$ in controls to $81.3 \pm 5.6\%$ after shA KD and $77.3 \pm 0.3\%$ after shB KD ($P = 0.0009$ and $P = 0.0005$, respectively). These data further suggest that *Kap1* and *Paupar* affect postnatal neurogenesis by disrupting both migration into the OB and morphological differentiation of newborn neurons.

## Discussion

LncRNAs can bind and regulate target genes on multiple chromosomes away from their sites of transcription. The number of lncRNAs that function in this way is steadily increasing suggesting that nuclear lncRNAs could exert a wide range of currently uncharacterised, *trans*-acting functions in transcription and chromatin regulation. Moreover, loss-of-function studies using animal model systems are needed to identify and characterise lncRNA regulatory roles during embryonic development and in adult tissue homeostasis to clarify the importance of this class of transcript *in vivo*.

To gain novel insights into lncRNA gene regulation, we investigated the mode of action of the CNS-expressed lncRNA *Paupar* at chromosomal binding sites away from its site of synthesis in N2A cells. We show that *Paupar* directly binds the KAP1 epigenetic regulatory protein and thereby regulates the expression of shared target genes important for proliferation and neuronal differentiation. Our data indicate that *Paupar* modulates histone H3K9me deposition at a subset of distal bound transcriptional regulatory elements through its association with KAP1, including at a binding site upstream of the *E2f2* gene. These chromatin changes are consistent with our previous report that this *E2f2* bound sequence functions as a transcriptional enhancer whose activity is restricted by *Paupar* transcript levels (Vance *et al*, 2014). Our results therefore suggest a model in which *Paupar*-directed histone modification changes in *trans* alter the activity of bound regulatory elements in a dose-dependent manner.

Several other lncRNAs have also been shown to alter the chromatin structure of target genes in *trans*. These include the human *PAUPAR* orthologue which can inhibit H3K4 trimethylation of the *Hes1* promoter in eye cancer cell lines, as well as lncRNA-HIT which induces p100/CBP-mediated changes in histone H3K27ac at bound sequences to regulate genes involved in chondrogenesis (Carlson *et al*, 2015; Ding *et al*, 2016). The lncRNA *Hotair* is one of the most studied *trans*-acting lncRNAs. Whilst *Hotair* has been proposed to guide PRC2 to specific locations in the genome to induce H3K27me3 and silence gene expression (Chu *et al*, 2011), recent conflicting studies report that PRC2 associates with low specificity to lncRNAs and suggest that *HOTAIR* does not directly recruit PRC2 to the genome to silence gene transcription (Kaneko *et al*, 2013; Davidovich *et al*, 2015; Portoso *et al*, 2017). Mechanistic studies on individual *trans*-acting lncRNAs such as *Paupar* are therefore needed to further define general principles of genome-wide lncRNA transcription and chromatin regulation.

It is proposed that lncRNAs may guide chromatin-modifying complexes to distal regions in the genome through RNA–RNA associations at transcribed loci, or either directly through RNA–DNA base pairing or indirectly through RNA–protein–DNA associations (Vance & Ponting, 2014; Rutenberg-Schoenberg *et al*, 2016). We show here that *Paupar* acts to increase KAP1 chromatin association by promoting the formation of a DNA binding regulatory complex containing *Paupar*, KAP1 and PAX6 within the regulatory regions of shared target genes in *trans*, as illustrated in the model in Fig 6. This suggests that *Paupar* functions as a cofactor for transcription factors such as PAX6 to modulate target gene expression across multiple chromosomes. In a similar manner, *Prncr1* and *Pcgem1* lncRNAs interact with the androgen receptor (AR) and associate with non-DNA binding cofactors to facilitate AR-mediated gene regulation (Yang *et al*, 2013). LncRNA-mediated recruitment of chromatin regulatory proteins to DNA bound transcription factors may represent a common mechanism of *trans*-acting lncRNA gene regulation, in line with their suggested role as molecular scaffolds (Tsai *et al*, 2010).

---

**Figure 5. *Paupar* and *Kap1* loss-of-function alters OB neuron number and morphology.**

P1 pups were electroporated with the indicated shRNA expression vectors. All shRNA plasmids also express GFP.

A, B   Immunostaining and quantification of GFP[+] cells that were electroporated in the SVZ and migrated to the OB, *Paupar* KD, 7 dpe. $N \geq 3$.

C, D   GFP[+] cells that have migrated to the olfactory bulb 7 dpe decrease after *Kap1* KD. Quantification of the density of electroporated cells in the OB after *Kap1* KD. $N = 3$.

E   High magnification showing different morphologies in GFP[+] granule layer OB neurons 7 dpe, *Paupar* KD. For ease of comparison, neuronal orientations were aligned to vertical. The cells shown in the scr control group are class I.

F   High magnification showing different morphologies in GFP[+] granule layer OB neurons 7 dpe, *Kap1* KD. Neuronal orientations rendered vertical. The scr control image shows several class I as well as class III neurons.

G   Quantification of the percentage of cells with Class I and Class III morphology 7 days after *Paupar* KD. $N \geq 3$.

H   Quantification of the percentage of cells with Class I and Class III morphology 7 days after *Kap1* KD. $N = 3$.

Data information: Data are shown as mean $\pm$ SEM and analysed by two-tailed Student's *t*-tests. *$P < 0.05$, **$P < 0.01$, ***$P < 0.001$. Scale bars represent 100 μm (A), 200 μm (C), 30 μm (E), 50 μm (F). The same scale is used for all images within each section.

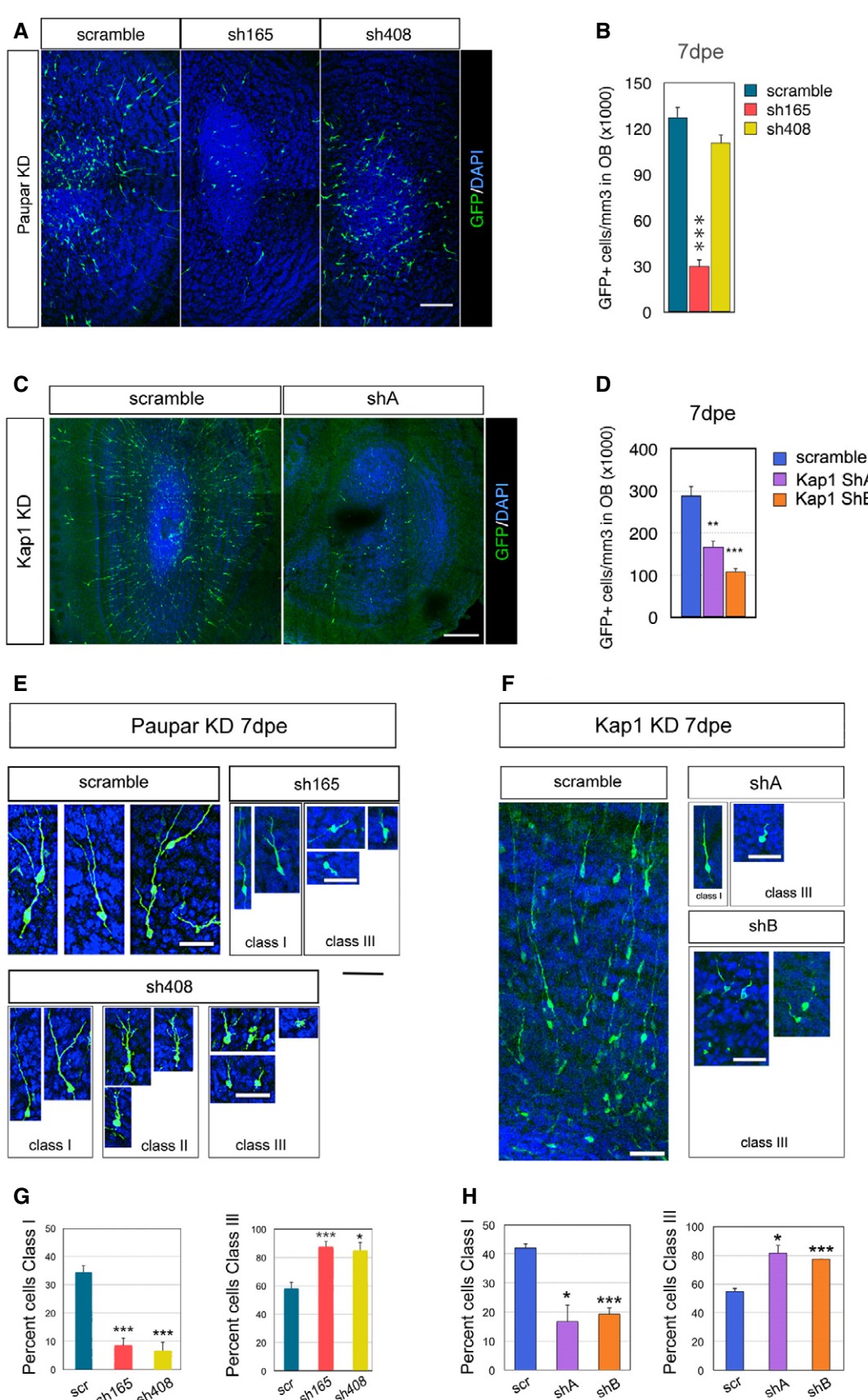

**Figure 5.**

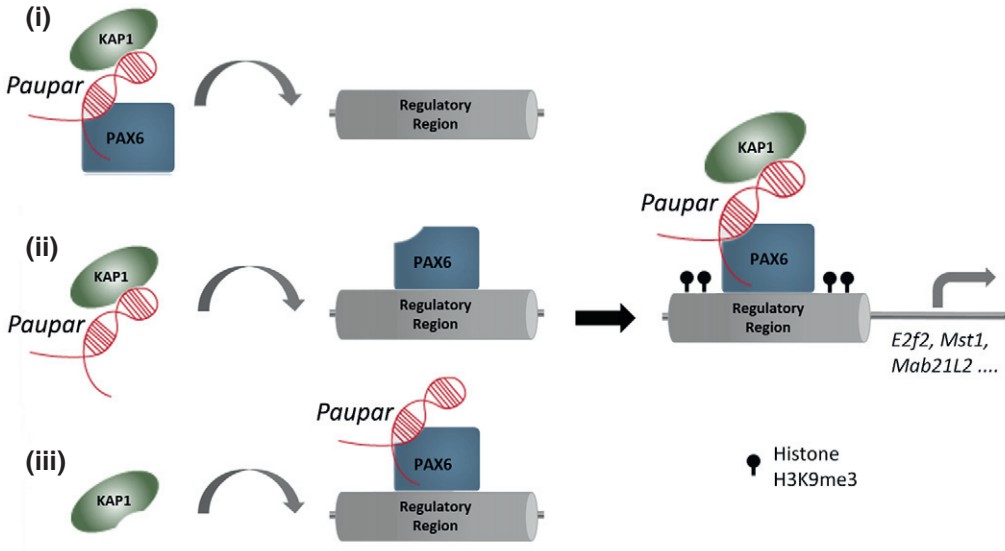

**Figure 6. Schematic detailing possible *Paupar* mode of action at distal bound regulatory regions.**

*Paupar* promotes KAP1 chromatin association and H3K9me3 deposition through the assembly of a DNA bound ribonucleoprotein complex containing *Paupar*, KAP1 and PAX6 within the regulatory regions of direct target genes such as *Mab21L2*, *Mst1* and *E2f2*. We propose three potential (non-mutually exclusive) scenarios to describe the order of assembly of this complex: (i) A ternary complex forms in the nucleoplasm before binding DNA; (ii) *Paupar* interacts with KAP1 and guides it to DNA bound PAX6; or (iii) KAP1 is recruited to a DNA bound PAX6-*Paupar* complex. This leads to local H3K9me3 modification changes at these bound sequences in *trans*. The model was generated taking into consideration the discovery that *Paupar* genome-wide binding sites contain an enrichment of motifs for neural transcription factors but are not enriched for sequences that are complementary to *Paupar* itself (Vance *et al*, 2014). This suggests that *Paupar* does not bind DNA directly but is targeted to chromatin indirectly through RNA–protein interactions with transcription factors such as PAX6. Moreover, KAP1 is a non-DNA binding chromatin regulator that is also targeted to the genome through interactions with transcription factors.

KAP1 is guided to the 3′ UTR of zinc finger genes in the genome through association with KRAB-ZNF transcription factors (O'Geen *et al*, 2007). However, the mechanisms of KAP1 genome-wide recruitment are not fully understood (Iyengar *et al*, 2011). Our data identify KAP1 as a novel RNA binding protein and show that *Paupar* plays a role in modulating the recruitment of KAP1 to specific PAX6 bound locations in the genome. We further assessed the extent to which *Paupar* may be able to modulate KAP1 genome-wide recruitment and identified 46 shared binding sites on chromatin, only one of which was within a 3′ UTR of a zinc finger gene. We measured PAX6 occupancy at a subset of these locations and identified significant PAX6 enrichment at 11 out of 15 *Paupar*-KAP1 co-occupied regions. These results raise the possibility that additional chromatin-enriched lncRNAs may operate to recruit KAP1 to specific locations in the genome and that this may involve context-specific interactions with both KRAB-ZNF and non-KRAB-ZNF containing transcription factors such as PAX6.

Our knockdown studies indicate that *Paupar* and *Kap1* are required for normal postnatal OB neurogenesis *in vivo*. At 7 dpe of the SVZ most cells would be migrating from the RMS into the OB with only a minority differentiating and our quantification supported this. In accordance with the KAP1-*Paupar* physical association, both *Paupar* and *Kap1* loss-of-function reduced the number of newborn neurons in the OB. Although the decrease in neuronal numbers could have been due to increased apoptosis caused by *Paupar* or KAP1 knockdown, we found no evidence of altered cell death in the SVZ, RMS or OB with TUNEL staining. Another possibility is that *Paupar* and KAP1 knockdown attenuated migration through the RMS. Reduced migration is sufficient to decrease rates of OB neurogenesis (Comte *et al*, 2011) and could explain the reduced number of cells in the OB 7 dpe. There is precedence for such an effect as reduction in either lncRNA *ncRNA-7a* or *HNF1A-AS1* decreased cell migration (Orom *et al*, 2010; Yang *et al*, 2014a), and reduced migration into the OB could also cause a delay in differentiation. Consistent with this, fewer newborn neuroblasts had differentiated morphology and more had immature morphology after *Paupar* or KAP1 depletion. Additionally, it may be that *Paupar* and KAP1 affect the rate of OB interneuron differentiation in a manner similar to NeuroD1 (Boutin *et al*, 2010; Pataskar *et al*, 2016).

This study identifies *Paupar* and *Kap1* as novel regulators of OB neurogenesis *in vivo* and provides important conceptual insights into the distal modes of lncRNA-mediated gene regulation. Given the widespread role played by *Kap1* in genome regulation and chromatin organisation, we anticipate that further chromatin-associated lncRNAs will be found to functionally interact with KAP1.

# Materials and Methods

### Plasmid construction

*Kap1* and *Paupar* targeting short hairpin RNAs (shRNAs), designed using the Whitehead Institute siRNA selection programme, were synthesised as double-stranded DNA oligonucleotides and ligated

into pBS-U6-CMVeGFP as shown previously (Vance *et al*, 2014). The *Paupar* targeting sh165 and sh408 expression constructs, the non-targeting scrambled control shRNA and pCAGGS-*Paupar* expression vector, are also detailed in (Vance *et al*, 2014). To generate the PAX6 expression vector, *Pax6* coding sequence was PCR amplified from mouse N2A cell cDNA as a NotI-XhoI fragment and inserted into pcDNA3.1(+) (Invitrogen). The forward primer incorporated a DNA sequence to insert the DYKDDDDK FLAG epitope tag in frame at the amino terminal end of PAX6. *Rcor3* coding sequence was also PCR amplified from mouse N2A cell cDNA and cloned into pcDNA3.1(+) to generate pcDNA3-RCOR3. pcDNA3-HA-KAP1 was a kind gift from Colin Goding (Ludwig Institute, Oxford). The sequences of the oligonucleotides used in this study are listed in Table EV1.

### Cell culture

N2A mouse neuroblastoma cells (ATCC CCL-131) were grown in DMEM supplemented with 10% foetal bovine serum. All transfections were performed using FuGENE 6 (Promega) following the manufacturer's instructions. To generate *Kap1* knockdown cells, $\sim 2 \times 10^5$ cells were plated per well in a six-well plate. 16–24 h later, cells were transfected with 1.5 μg *Kap1* shRNA expression construct and 300 ng (5:1 ratio) pTK-Hyg (Clontech). Three days after transfection, cells were trypsinised, resuspended in growth medium containing 200 μg/ml Hygromycin B and plated onto a 6-cm dish. Drug-resistant cells were grown for 7 days and harvested as a pool.

### Immunoprecipitation

$1 \times 10^6$ N2A cells were seeded per 10-cm dish. The next day, cells were transfected with different combinations of pcDNA3-FLAG-PAX6, pcDNA3-HA-KAP1, pcDNA3-RCOR3, pCAGGS-Paupar, pCAGGS-AK034351 control transcript or pcDNA3.1 empty vector. 6 μg plasmid DNA was transfected in total. Two days later, cells were washed twice with ice-cold PBS, transferred to 1.5-ml microcentrifuge tubes and lysed in 1 ml ice-cold IP Buffer (IPB; 50 mM Hepes pH 7.5, 350 mM NaCl, 1 mM MgCl$_2$, 0.5 mM EDTA and 0.4% IGEPAL CA-630) for 30 min, 4°C with rotation. Lysates were pelleted at 16,000 *g*, 20 min, 4°C in a microfuge, supernatant was added to 30 μl anti-FLAG M2 Magnetic Beads (#M8823, Sigma) and incubated overnight at 4°C with rotation. Beads were washed three times with IPB and eluted in 20 μl Laemmli sample buffer for 5 min at 95°C. Bound proteins were detected by Western blotting using anti-FLAG M2 (F3165, Sigma), anti-KAP1 (ab10483, Abcam), anti-RCOR3 (A301-273A, Bethyl Laboratories) and Protein A HRP (ab7456, Abcam).

### RNA pulldown assay

Sense RNA was *in vitro*-transcribed from pCR4-TOPO-*Paupar* using T7 RNA polymerase, according to manufacturer's instructions (New England Biolabs). Transcribed RNA was concentrated and purified using the RNeasy MinElute Cleanup kit (Qiagen). Purified RNA was then 5′ end labelled with biotin-maleimide using a 5′ EndTag nucleic acid labelling system (Vector Laboratories). Streptavidin-coated Dynabeads M-280 (Invitrogen) were washed, prepared for RNA

manipulation and the 5′ biotinylated RNA bound according to manufacturer's instructions. N2A cell nuclear extract was diluted in affinity binding/washing buffer (150 mM NaCl, 50 mM HEPES, pH 8.0, 0.5% Igepal, 10 mM MgCl$_2$) in the presence of 100 μg/ml tRNA, 40 U/ml RNaseOUT (Invitrogen) and a protease inhibitor cocktail (Roche). RNA-coated beads were incubated with nuclear extract at room temperature for 2 h with rotation. The supernatant was then removed, the beads washed six times (10 min) with affinity/binding washing buffer, and bound protein eluted by heating to 95°C in the presence of Laemmli sample buffer for 5 min. Samples were loaded onto a 10% Tris-glycine polyacrylamide gel (Bio-Rad) and subjected to denaturing SDS–PAGE until they just entered the resolving gel. Protein samples were then excised, diced and washed three times with nanopure water. Tryptic digest and mass spectrometry were performed by the Central Proteomics Facility (Dunn School of Pathology, University of Oxford).

### RNA-IP

Approximately $1 \times 10^7$ N2A cells were used per RNA-IP. Native RNA-IP experiments were performed using the Magna RIP Kit (Millipore) according to the manufacturer's instructions. UV-RIP was carried out as described in Vance *et al* (2014). We used the following rabbit polyclonal antibodies: anti-RCOR3 (A301-273A, Bethyl Laboratories), anti-CoREST (07-455, Millipore), anti-KAP1 (ab10483, Abcam), anti-ERH (ab96130, Abcam), anti-PPAN (11006-1-AP, Proteintech Group) and rabbit IgG (PP64B, Millipore).

### Chromatin immunoprecipitation

For knockdown experiments, $4 \times 10^6$ N2A cells per ChIP were seeded in 15-cm plates. The next day, cells were transfected with either 15 μg *Paupar* targeting shRNA expression vectors or a non-targeting scr control. Three days later, cells were harvested for ChIP using either 5 μg anti-KAP1 (ab10483, Abcam), anti-histone H3K9me3 (39161, Active Motif) or normal rabbit control IgG (#2729, Cell Signalling Technology) antibodies. ChIP was performed as described in Vance *et al* (2014). 5 μg anti-PAX6 (#AB2237, Millipore) was used for PAX6 ChIP. For KAP1 ChIP-seq, the following modifications were made to the protocol: $\sim 2 \times 10^7$ N2A cells per ChIP were double-cross-linked, first using 2 mM disuccinimidyl glutarate (DSG) for 45 min at room temperature, followed by 1% formaldehyde for 15 min at room temperature, as described in Nowak *et al* (2005). Chromatin was sheared to ~200 bp using a Bioruptor Pico (Diagenode) and ChIP DNA and matched input DNA from two independent KAP1 ChIP experiments were sequenced on an Illumina HiSeq 4000 (150-bp paired-end sequencing).

### ChIP-seq analysis

The Babraham Bioinformatics *fastqscreen* (https://www.bioinformatics.babraham.ac.uk/projects/fastq_screen/) and *fastQC* (https://www.bioinformatics.babraham.ac.uk/projects/fastqc/) tools were used to screen the raw reads for containments and to assess quality. We removed traces of the adapter sequence from the raw reads using the *Trimmomatic* tool (Bolger *et al*, 2014). Trimmomatic was also used to trim by quality with the options: *LEADING:3 TRAILING:3 SLIDINGWINDOW:4:15 MINLEN:50*. The trimmed reads

were aligned to the mm10 reference genome, using the Burrows-Wheeler Aligner (Li & Durbin, 2010) with the command: > *bwa mem mm10 < pair_1.fq > < pair_2.fq >*. Alignment quality was assessed with the *Qualimap 2.2.1* tool (Okonechnikov *et al*, 2016). The aligned reads were filtered to exclude reads with a MAPQ alignment quality < 20. Furthermore, we excluded reads aligning to blacklisted regions identified by the ENCODE consortium (ENCODE Project Consortium, 2012). MACS2 version 2.1.1.20160309 was used to identify genomic regions bound by KAP1. We further filtered the aligned reads to retain only those with length 150 and called peaks relative to the input controls using the options "*–gsize = 1.87e9 –qvalue = 0.01 -B –keep-dup auto*". To examine the read density distribution in the vicinity of KAP1 peaks, we used *deepTools* (Ramirez *et al*, 2016). Read density was calculated with respect to input using the bamCompare tool from deepTools, with the option "*–binSize 10*". The matrix of read densities in the vicinity of KAP1 peaks was calculated using "*computeMatrix reference-point*", and heatmaps plotted with "*plotHeatmap*". The Genomic Association Test tool *GAT* (Heger *et al*, 2013) was used to characterise KAP1 binding sites and the relationship between KAP1 and *Paupar*. Coordinates with respect to the mm10 reference genome for characteristic genomic regions (exons, introns, 3′ UTRs, etc.) were downloaded from the UCSC Genome Table Browser (https://genome.ucsc.edu/cgi-bin/hgTables). The enrichment of KAP1 peaks and the intersection of KAP1 and *Paupar* peaks with respect to these genomic regions was assessed using GAT with the options "*–ignore-segment-track –num-samples = 100,000*" and using the complement of the blacklist regions as the workspace. To test for significance coincidence of KAP1 and *Paupar* peaks, we use GAT with the same options. The *Paupar* CHART-Seq peakset from Vance *et al* (2014) was used for comparison.

## Transcriptomic analysis

Total RNA was isolated from triplicate control and KAP1 knock-down cells using the Qiagen Mini RNeasy kit following the manufacturer's instructions. RNA samples with a RNA Integrity Number greater than 8, as assessed on a BioAnalyzer (Agilent Technologies), were hybridised to Mouse Gene 1.0 ST Arrays as detailed in (Chalei *et al*, 2014). Microarray data were Robust Multi-array Average (RMA) normalised using GeneSpring GX12.6 (Agilent). Differentially expressed genes (fold change difference ≥ 1.4) were identified using a false discovery rate of ≤ 0.05 with a Benjamini and Hochberg multiple testing correction (Limma). Gene Ontology analysis was performed as previously (Vance *et al*, 2014).

## Neurosphere assay

Neurospheres were cultured according to standard protocols as previously described (Dizon *et al*, 2006). In brief, age P3-P6 CD1 mice pups were anesthetised by hypothermia and decapitated, and the brains were immediately dissected out and sectioned in the coronal plane with a McIlwain tissue chopper. The SVZ was then dissected out in ice-cold HBSS in a sterile laminar flow hood. Accutase was used for 15 min for cell dissociation. Cells were cultured in defined Neurobasal media supplemented with 20 ng/ml EGF (Sigma) and 20 ng/ml bFGF (R&D). Cells were seeded at a density of 100 cells/μl and passaged every 3–4 days.

## Neural stem cell nucleofection

$3–4 \times 10^6$ dissociated neurosphere cells were nucleofected according to the protocol of LONZA (VPG-1004). Cells were mixed with 100 μl nucleofection solution (82 μl of Nucleofector Solution + 18 μl of supplement) and 5–10 μg DNA and transferred into cuvettes. 500 μl of culture medium was added into the cuvette, and the sample was then transferred into 1 ml medium and centrifuged at 250 *g* for 5 min. Cells were resuspended with fresh medium and plated at 200,000 cells/2 ml in a PolyHeme-coated 6-well plate.

## Postnatal electroporation

Electroporation was performed as published (Boutin *et al*, 2008; Chesler *et al*, 2008). DNA plasmids were prepared with Endofree Maxi kit (Qiagen) and mixed with 0.1% fast green for tracing. DNA concentrations were matched in every individual experiment. P1 CD1 pups were anesthetised with hypothermia, and 1–2 μl of plasmids was injected with glass capillary. Electrical pulses (100 V, 50 ms ON with 850-ms intervals for five cycles) were given with tweezer electrodes (CUY650P5). Pups were recovered, then returned to dam and analysed at the indicated time.

## Immunohistochemistry and imaging

Immunohistochemistry was as previously described (Young *et al*, 2014) using Chicken anti-GFP (1:500, Aves) and goat anti-DCX (1:100, Santa Cruz) primary and Alexafluor-conjugated (Invitrogen) secondary antibodies. TUNEL method was performed using the *In situ* cell death detection kit, TMR red (Roche-12156792910) to detect apoptosis. Sections were imaged with Zeiss 710 Laser Scanning Microscopy. For co-localisation in GFP$^+$ cells, a 20× or 40× oil immersion objective was used and Z stacks were generated at 2-μm intervals. Confocal images were analysed with ImageJ.

## Morphological evaluation

All GFP$^+$ neuroblasts in the granule layer of the OB were binned into Class I, II or III groups similar to a previous study (Boutin *et al*, 2010). Only cells with obvious cell bodies and that were entirely found in the field were included. Cells in the rostral migratory stream in the core of the OB, and in OB layers outside of the granule layer, were not included. *N* = 3–5 mice per group.

## Ethics

All mouse experiments were performed in accordance with institutional and national guidelines and regulations under UK Home Office Project Licence PPL 3003311.

## Data availability

The ChIP-Seq and microarray data have been deposited in the GEO database https://www.ncbi.nlm.nih.gov/geo/) under the following accession numbers: GSE110032 and GSE110033.

Expanded View for this article is available online.

## Acknowledgements

This project has been funded by a Biotechnology and Biological Sciences Research Council grant to KWV (BB/N005856/1; KWV, IP), a Medical Research Council (MR/M010554/1; FGS, BS, FA) grant to FGS, and European Research Council (Project Reference 249869, DARCGENs), Medical Research Council (MC_UU_12008/1; CPP, NC) and Wellcome Trust (106956/Z/15/Z; CPP, TS) grants to CPP.

## Author contributions

KWV conceived the study. IP, FA, BS, TS, SL, FGS and KWV designed and performed the experiments. IP, FGS and KWV analysed and interpreted the data. NC, SL and DJW carried out computational analysis of the microarray and ChIP-seq data. KWV and FGS wrote the manuscript with input from IP and CPP who reviewed and edited the drafts. CPP, FGS and KWV supervised the research and acquired funding.

## Conflict of interest

The authors declare that they have no conflict of interest.

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
