## [Review Process File · The EMBO Journal]

***Paupar* LncRNA Promotes KAP1 Dependent Chromatin Changes And Regulates Olfactory Bulb Neurogenesis**

Ioanna Pavlaki, Farah Alammari, Bin Sun, Neil Clark, Tamara Sirey, Sheena Lee, Dan J Woodcock, Chris P Ponting, Francis G Szele and Keith W Vance.

Review timeline:

Submission date:	2 nd September 2016
Editorial Decision:	4 th October 2016
Authors' response:	1 st November 2016
Editorial response:	2 nd November 2016
Revision received:	12 th September 2017
Editorial Decision:	21 st November 2017
Revision received:	22 nd January 2018
Editorial Decision:	22 nd February 2018
Revision received:	5 th March 2018
Accepted:	8 th March 2018

Editor: Anne Nielsen

Transaction Report:

1st Editorial Decision

4th October 2016

Thank you for submitting your manuscript for consideration by The EMBO Journal. It has now been seen by three referees whose comments are shown below. As you will see, while the referees all express interest in the work and topic in principle, they do not offer strong support for publication in The EMBO Journal - at least at this stage of analysis.

I will not repeat all their individual points of criticism here, but it becomes clear that our referees are concerned that the depth of analysis is too limited and that the study is thus too premature for them to support its publication. More specifically, you will see that ref #1 asks for extensive, additional work on the neurogenesis part as well as insight on the chromatin status of the affected target genes. The latter point is shared by ref #3 who furthermore lists a number of controls for specificity, causality and functional contribution that would have to be included. Clearly, an extensive amount of further experimentation would be required to address these issues and to bring the study to the level of insight and significance required for publication here. Furthermore, the outcome of such experiments cannot be predicted at this point and would thus lie outside the scope and the timeframe of a revision. I therefore see little choice but to come to the conclusion that we cannot offer to publish the manuscript at this point. If you were to extensively revise the manuscript along the lines outlined by the referees - and if the original conclusions still hold true - we could be willing to look at the study again at a later stage, although this would have to be as a new independent submission.

Given the overall negative opinions from the referees, I am afraid we are unable to offer further steps towards publication in The EMBO Journal at this stage.

REFEREE REPORTS

Referee #1:

Sun et al investigate the role of the lncRNA Paupar in brain development. Previous studies have linked this transcript to regulation of Pax6 expression and activity and the authors now investigate how. Both in vivo and in vitro experiments show that knockdown of Paupar impairs neurogenesis by binding the chromatin regulatory factor Kap1. A number of experiments are performed supporting a model in which Paupar/Kap1/Pax6 interact to control the expression of genes involved in neurogenesis.

This work is insightful, novel and provides a better understanding of a poorly studied class of transcripts. Some points are however vaguely addressed, which I suggest the authors to consider for strengthening their work.

Major points

1) It is emphasized that Kap1 is a "key epigenetic regulator" and that this underlies its function as Pax6/Paupar cofactor. Yet, no effect in epigenetic marks is investigated. These are not difficult to assess in vitro or even better in vivo. For example, electroporated cells could be FAC-sorted and bisulphite, ATAC, ChIP or else performed followed by qPCR of the 3-4 target loci identified by the authors (E2f2; Mst1; etc). This could validate that epigenetic marks (DNA methylation, chromatin and/or histones) in these loci are the result of Paupar manipulation. This would be an important and major addition.

2) The assessment of neurogenesis is rather superficial and confusing....there are many issues here, some partly redundant, and a couple of experiments can take care of most of them:

i) The authors realize that an increase in neuroblasts contradicts a decrease in neurons and argue that this is due to depletion of stem cells. But this is very unlikely! These data (Fig 1I-J) were obtained 7 days post electroporation and these neurons are the ones generated and migrating to the OB before there could have been depletion since migration itself requires 1+ week. A decrease in neurons due to stem cell depletion cannot become apparent unless a longer time is considered, 2-3 or more.

ii) The caspase in Fig S1C is not sufficient to exclude apoptosis (which may explain the above contradiction). This is because S1C only looks at the SVZ and plenty of neurons may be dying in the RMS/OB. Also, apoptosis may appear and be cumulative over different periods of time while this analysis only looks at 3 days.

iii) The assessment of stem, progenitors and neuroblasts is superficial. More markers are needed, such as Egfr, Dcx or more in combination with Mash. Effects on quiescence would also represent a nice addition by long-term label retention with BrdU.

iv) The statement of fewer cells in Fig S1C needs statistical assessment. This reduction is hard to explain given the short period of time (24h). If, for whatever reason, Paupar constructs target less cells then little wonder there are also less neurons (point above).

v) Any serious assessment of neurogenesis in vivo requires some basic stereology.

vi) Neurosphere cultures are also superficially assessed. Proliferating versus differentiation conditions are not compared and multipotency neither.

3) I don't understand in Fig 2F why there is a band in lane Kap1/WB:Kap1 after Flag-IP. Kap1 is not being flagged, so it should not be IP. This lane contains no Flag-Pax6, so there should be no association...

4) The "relative expression" of Paupar in vitro and vivo shown in Fig 1A-B does not tell anything given that this is compared to non-RT. The authors should compare it to other genes as well as showing it in tissue by ISH. To their help identification and functional relevance of lncRNAs in the

developing mouse brain has been done by many groups including at the level of individual cell types in Aprea et al (couple of papers....that surprisingly were missed by the authors in the reference list). The authors can simply use these publicly available resources to show Paupar expression.

Referee #2:

General comments

The manuscript submitted by Sun et al follows up on work previously published by the group on the role of a long non-coding RNA, Paupar, in regulating neurogenesis. Their previous work (Vance et al, EMBO J. 2014) used an in vitro system to establish that Paupar lncRNA physically interacts with Pax6 transcription factor to co-regulate gene expression in a neuroblastoma line. Here they sought to extend the understanding of Pauper's role in neurogenesis by perturbing its expression in vivo and further evaluating its biochemical partnerships in vitro. In vivo, they describe alterations in the subventricular zone (SVZ) after electroporating Pauper shRNA. In vitro, they go on to show that Pauper binds to the KAP1 chromatin regulatory protein and examine the role of Pauper-KAP1 on gene expression, genome-wide. These two approaches combine to make a modest advance on our understanding of the role of Paupar in regulating neurogenesis.

Overall the experiments were executed and interpreted carefully, the data were presented clearly. The writing was generally of good quality although some modifications are suggested below to improve the presentation.

The in vivo work is mostly descriptive and does make the case that knocking down Paupar expression disrupts neurogenic pathways in SVZ. However, it is unclear from this line of investigation if its role is in maintenance of the stem cell niche or is its primary role in controlling early neural development. Their work sheds light on Paupar's importance, but a more detailed understanding of its biological role will require further studies.

The in vitro work is quite solid. They have made a strong case for the formation of a regulatory complex that comprises Pax6-Pauper-KAP1. They show that Paupar is important in directing Pax6 and KAP1 to genomic locations. This is another example of a role for a long non-coding RNA as key component of a gene regulatory network that acts as a scaffold that aids in assembly of transcriptional machinery.

Specific comments

1. The introduction spends too much effort on the general importance of lncRNAs and not enough on the background of Paupar (and Pax6).
2. Adhere to convention in Results to use past tense in describing results. See page 9, para 2 for examples, but this should be corrected throughout.
3. Also need to adhere to convention on gene names. Sometimes they use 'PAX6' and others it is 'Pax6'. All this work was done in mouse, so lower case and italics for gene names is appropriate.
4. P6, para 2: "P4" should be spelled out as "passage 4"
5. P6, para 2: for clarity, change "N2A cells" to "mouse N2A neuroblastoma cells"
6. P6, para 3: The second sentence is quite vague here. But the topic is better handled later (p7, para 3). It needs revision or just wait till the subsequent section to describe that GFP is marking transfected cells, and explain why the majority of cells are radial glial-like.
7. P7, para1: no need to spell out "hpe" again
8. P8, para 3: "native RNA-IP experiments" is jargon. Needs to spell out for general readership.
9. P11 para 2: "CHART-seq mapped" is jargon and the topic is presented too vaguely here. Consider addressing the CHART-seq information in the introduction.

10. Methods: for clarity state that CD1 pups are mice.

Referee #3:

In this work, Sun and colleagues describe the identification of a novel ribonucleoprotein complex in neuroblastoma N2A cells comprising the lnc RNA Paupar, KAP1 and PAX6, the deposition of which at target genes is promoted by Paupar in trans, with impact on neural stem cell self-renewal and differentiation.

However, the demonstration suffers from many shortcomings and the proposed model is very premature. In particular, the Paupar-KAP1 connection needs major strengthening through more advanced chromatin and biochemical analyses, and the acquisition of data that address more directly KAP1 function in this complex, including through the use of previously characterized KAP1 mutants.

Other, specific comments:

1. Figure 1:

- Fig. 1a,b: without controls (other transcripts, normalisation, other tissues, testing NS and SVZ in same experiment) these data are not informative. What is the error bar? Why is it absent in b? The impact on Paupar should be quantified, not just assumed (particularly relevant since invoking dose-dependence). Figure 1a, 1b are missing statistics and ticks on Y axis (present otherwise in Figures 3 and 4).

- Fig 1c is not useful.

- Fig. 1d: what does "electroporated GFP+ cells" mean? Nowhere is it specified the nature of the shRNA vectors, and whether they express GFP. Ki67 staining is invisible.

- Fig. 1e: what does " $N \geq 3$ " mean? What techniques was used to determine that a GFP+ cell expressed or not the indicated markers?

- Fig. 1f: $N=?$

- Fig. 1h: same as 1e. Where are "the small arrows"?

- Was it impossible to verify the specificity of the Paupar KD via complementation?

2. Figure 2:

- 2c: RNA IP experiments are notoriously subjected to false positive. KAP1 mutant proteins could serve as useful controls.

- 2f: KAP1 alone is brought down by the flag antibody, indicating lack of specificity under the conditions used here. The PAX6 input lane should be shown. An IgG control should be used.

- Figure 2c, d, e missing statistics and ticks on Y axis;

3. Figure 3:

- Complementation with WT or mutant KAP1 would be nice.

- Without KAP1 ChIP-Seq data in these cells, the finding of up- and down-regulated genes upon KAP1 KD does not primarily suggest that KAP1 can function as either an activator or a repressor, but rather a superimposition of primary and secondary effects.

- Why do the authors define the shRNA control as "scramble" in Figure 1 and "sh Cont" in Figure 3? Even though different scramble sequences might have been used, this is confusing.

- Figure 3b is missing statistics and Y axis ticks.

- Figure 3c: A fold-change of 1.4 is really low; most investigators will use 2 as minimal threshold for differential gene expression.

4. Figure 4: Some positive controls should be used for the KAP1 ChIP-PCR (e.g. the 3' end of KRAB-ZFP genes, so as to get a better appreciation of the relative enrichment at tested loci.

5. Figure 5: This control becomes even more relevant here. Sites need to be explored, where KAP1 recruitment is Paupar-independent. The only one here is Ezh2BS, which is Paupar+ by CHART... A ChIP-Seq would be far more valuable than the presented ChIP-PCR, as it would give a global view of KAP1 recruitment in Paupar-high and Paupar-low conditions, and also performed following Pax6 manipulations.

- Figure 5a, an error bar (presumably Scr) is out of the chart.

- Figure 5d and 2f are in clear conflict. In figure 5d following FLAG-PAX6 transfection the authors detect KAP1 in pull-down complexes but the same was not observed in Figure 2f (top panel, second lane). Why do they overexpress exogenous KAP1 whereas this protein is already present at good levels in N2A cells?

6. Discussion:

- The authors cannot claim that KAP1 interacts directly with Paupar, since they did not perform

binding assays with purified recombinant protein and RNA.

- More generally, the authors say very about the functional role of KAP1, in a putative Paupar-Pax6-KAP1 complex. If they think it acts as a coactivator owing to some post-translational modification, they should test this (probing for pS473 and pS824 KAP1 by ChIP, co-IP, etc.).

- The recruitment of KAP1 to the 3' end of zinc finger genes is KRAB-mediated (a statement on page 16 suggests otherwise).

Authors' Response

1st November 2016

Thank you very much for sending through the referees' comments for our manuscript (Ref: EMBOJ-2016-95650). We appreciate the comments raised by the reviewers and have found them to be very helpful. Several of the comments can be addressed immediately whilst we are preparing additional experiments to address the remainder. The results of these will further characterise the function of Paupar in neurogenesis and chromatin regulation and we hope to resubmit this study next year if the original conclusion still hold true.

Editorial Response

2nd November 2016

Thanks for contacting me about this. I am glad to hear that the referee comments are helpful to you and that you will be able to address them experimentally. I look forward to seeing the new version of the manuscript next year, assuming that things work out as planned.

1st Revision - authors' response

12th September 2017

Referee #1:

Sun et al investigate the role of the lncRNA Paupar in brain development. Previous studies have linked this transcript to regulation of Pax6 expression and activity and the authors now investigate how. Both in vivo and in vitro experiments show that knockdown of Paupar impairs neurogenesis by binding the chromatin regulatory factor Kap1. A number of experiments are performed supporting a model in which Paupar/Kap1/Pax6 interact to control the expression of genes involved in neurogenesis.

This work is insightful, novel and provides a better understanding of a poorly studied class of transcripts. Some points are however vaguely addressed, which I suggest the authors to consider for strengthening their work.

We thank the reviewer for the positive critique of our study. We have now substantially strengthened our study by performing additional experiments which we describe below and in the considerably revised manuscript.

Major points

1) It is emphasized that Kap1 is a "key epigenetic regulator" and that this underlies its function as Pax6/Paupar cofactor. Yet, no effect in epigenetic marks is investigated. These are not difficult to assess in vitro or even better in vivo. For example, electroporated cells could be FAC-sorted and bisulphite, ATAC, ChIP or else performed followed by qPCR of the 3-4 target loci identified by the authors (E2f2; Mst1; etc). This could validate that epigenetic marks (DNA methylation, chromatin and/or histones) in these loci are the result of Paupar manipulation. This would be an important and major addition.

To identify Paupar-mediated chromatin changes at bound loci we depleted Paupar expression using shRNA transfection in N2A cells and performed ChIP-qPCR, as suggested. This measured the levels of histone H3K9me3 (a mark that is deposited by KAP1 through its interaction with the SETDB1 methyltransferase) at shared bindings sites within *Mab21L2*, *Mst1*, *E2f2* and *Igfbp5*, genes which are co-regulated by Paupar, Kap1 and Pax6. This identified a significant reduction in H3K9me3 deposition for 3 of 4 of these regions. These changes are consistent with reduced KAP1 chromatin occupancy and together indicate that Paupar is able to promote KAP1-dependent chromatin changes at distal bound sequences.

2) *The assessment of neurogenesis is rather superficial and confusing....there are many issues here, some partly redundant, and a couple of experiments can take care of most of them:*

In summary, we have now performed the following experiments that have further investigated the effect of both *Paupar* and *Kap1* on neurogenesis:

- We have added an analysis of *Dlx* expression in the subventricular zone (SVZ). As stated in the Discussion, in addition to being a marker of TAPs and neuroblasts, *Dlx* drives neurogenesis.
- We have added an analysis of *Kap1* knockdown in the SVZ and relate it to *Paupar* knockdown.
- We also have added a new analysis of both *Paupar* and *Kap1* knockdown effects in the olfactory bulb (OB), the final target for SVZ neurogenesis.
- We controlled for apoptosis and show that the number of *Caspase 3* or TUNEL+ cells does not change in the SVZ, rostral migratory stream (RMS) or olfactory bulbs.

i) The authors realize that an increase in neuroblasts contradicts a decrease in neurons and argue that this is due to depletion of stem cells. But this is very unlikely! These data (Fig 11-J) were obtained 7 days post electroporation and these neurons are the ones generated and migrating to the OB before there could have been depletion since migration itself requires 1+ week. A decrease in neurons due to stem cell depletion cannot become apparent unless a longer time is considered, 2-3 or more.

We agree with this assessment and thus have removed the notion of stem cell depletion from the text. Importantly our new loss-of-function experiments with *Kap1* (Fig 6d) also show a reduction in the number of newborn neurons that have migrated into the OB.

ii) The caspase in Fig S1C is not sufficient to exclude apoptosis (which may explain the above contradiction). This is because S1C only looks at the SVZ and plenty of neurons may be dying in the RMS/OB. Also, apoptosis may appear and be cumulative over different periods of time while this analysis only looks at 3 days.

We agree. In response, we conducted extensive new experiments on apoptosis to complement our previous data on *Caspase 3* expression. We used TUNEL assays showing there were no differences among scr, sh165 and sh408 groups in the SVZ, RMS and the OB at 24hpe, 3dpe and 7dpe after *Paupar* knockdown. The results in supplemental Fig S4b, c show immunostaining and quantification of the percentage of GFP+ cells that are TUNEL+ in the SVZ at 3dpe as a representative example. Similarly, knockdown of *Kap1* by ShA and ShB did not alter the percentage of electroporated GFP+ cells that were TUNEL+ (Supplemental Fig S4d, e).

*iii) The assessment of stem, progenitors and neuroblasts is superficial. More markers are needed, such as *Egfr*, *Dcx* or more in combination with *Mash*. Effects on quiescence would also represent a nice addition by long-term label retention with *BrdU*.*

In response, we now include DLX2 immunohistochemistry and quantification both after *Paupar* and *Kap1* knockdown. In summary, we now show 4 markers: *GFAP*, *Mash1*, *Dlx2* and *Ki67*; importantly, *Mash1* and *Dlx2* are not simply inert markers but are necessary for neurogenesis. We agreed that further examination of neuroblasts is important. Consequently, we added *Dcx* immunohistochemistry to the study (Supplemental Fig 3d). In line with this we also performed a study of newborn neurons in the OB, the final target of the SVZ neurogenic niche. We present novel data showing that *Kap1* knockdown diminished the number of newborn neurons and that both *Paupar* and *Kap1* loss caused abnormal neuroblast morphology in the OB (Fig 6). The *BrdU* experiments suggested for long-term label retaining cells would indeed be informative but would necessitate multiple injections which, combined with neonatal electroporation, would be hard to justify to UK animal use authorities. We considered adding *BrdU* to the drinking water but whereas this is viable in adults, is not reproducible in neonates.

*iv) The statement of fewer cells in Fig S1C needs statistical assessment. This reduction is hard to explain given the short period of time (24h). If, for whatever reason, *Paupar* constructs target less cells then little wonder there are also less neurons (point above).*

To address this, we quantified the numbers of GFP+ cells at both 24hpe and 3dpe in sh165 and sh408 groups. Indeed, we observed a statistically significant decrease in the number of GFP+ cells in the SVZ, but only in the sh165 group. We have never observed this previously in other electroporations despite having performed many different labelling, loss-of-function or gain-of-function studies with this technique. In fact, quantification of the number of GFP+ cells in the SVZ at 24hpe and 3dpe after sh408 did not reveal differences in cell numbers. Similarly, the number of GFP+ cells at 3dpe in both *Kap1* KD groups was not changed from Scr controls. We note that co-electroporation of a different labelling construct with sh165 also resulted in fewer cells being labelled (Supplemental Fig 3c). To control for the differences in electroporation efficiency in sh165, we systematically counted the percentage of GFP+ cells that expressed various markers.

v) *Any serious assessment of neurogenesis in vivo requires some basic stereology.*

We agree with this statement with regards to the olfactory bulb and express the numbers of GFP+ cells in the OB stereologically. Stereology relies on 3 fundamental requirements for accurate assessment of absolute numbers of cells: round to oval shape of the brain region being quantified, homogeneous distribution of cells and ideally thick >50 micron tissue sections. We were interested in the differences in cell numbers between controls and *Paupar* or *Kap1* knockdown rather than in absolute numbers. As well, the SVZ has a very thin and flat shape. SVZ cells are not distributed homogeneously but are clustered in multiple complex arrangements. Thus we do not favour stereological assessments in the SVZ. Note also, for the reasons described above we are expressing results as the percentage of cells expressing markers, rendering stereology unnecessary.

vi) *Neurosphere cultures are also superficially assessed. Proliferating versus differentiation conditions are not compared and multipotency neither.*

We agree and have removed the neurosphere data. Although neurospheres can be an important adjunct in SVZ research the main point of studying the SVZ was to use it as an *in vivo* exemplar of *Paupar* and *Kap1* function, rendering neurospheres less useful.

3) *I don't understand in Fig 2F why there is a band in lane Kap1/WB:Kap1 after Flag-IP. Kap1 is not being flagged, so it should not be IP. This lane contains no Flag-Pax6, so there should be no association...*

This band indicates that a small amount of transfected KAP1 non-specifically binds to the FLAG beads. We tried increasing the salt concentration in the IP buffer as well as salt and detergent in the wash buffer to reduce binding of KAP1 to beads alone but were unable to remove this completely. Despite this, Fig 1f (Fig 2f in the previous manuscript) clearly shows an enrichment of KAP1 binding in complex with transfected FLAG-PAX6 (lane 4) compared to the KAP1 alone transfection (lane 3).

4) *The "relative expression" of Paupar in vitro and vivo shown in Fig 1A-B does not tell anything given that this is compared to non-RT. The authors should compare it to other genes as well as showing it in tissue by ISH. To their help identification and functional relevance of lncRNAs in the developing mouse brain has been done by many groups including at the level of individual cell types in Aprea et al (couple of papers....that surprisingly were missed by the authors in the reference list). The authors can simply use these publicly available resources to show Paupar expression.*

In response, we now present *Paupar* expression relative to the housekeeping gene *Gapdh*. This has enabled us to compare *Paupar* expression in the SVZ, neurospheres and N2A cells (Supplemental Fig 3a). The *in situ* hybridization was a good idea. We tried several times to optimise FISH in SVZ cells, however the probes assayed were unreliable. We appreciate the excellent work of Aprea, Calagari and colleagues yet would like to note that these were in developing mouse cortex not in the postnatal SVZ.

Referee #2:

General comments

The manuscript submitted by Sun et al follows up on work previously published by the group on the role of a long non-coding RNA, Paupar, in regulating neurogenesis. Their previous work (Vance et al, EMBO J. 2014) used an in vitro system to establish that Paupar lncRNA physically interacts with

Pax6 transcription factor to co-regulate gene expression in a neuroblastoma line. Here they sought to extend the understanding of Pauper's role in neurogenesis by perturbing its expression in vivo and further evaluating its biochemical partnerships in vitro. In vivo, they describe alterations in the subventricular zone (SVZ) after electroporating Pauper shRNA. In vitro, they go on to show that Pauper binds to the KAP1 chromatin regulatory protein and examine the role of Pauper-KAP1 on gene expression, genome-wide. These two approaches combine to make a modest advance on our understanding of the role of Pauper in regulating neurogenesis.

Overall the experiments were executed and interpreted carefully, the data were presented clearly. The writing was generally of good quality although some modifications are suggested below to improve the presentation.

The in vivo work is mostly descriptive and does make the case that knocking down Pauper expression disrupts neurogenic pathways in SVZ. However, it is unclear from this line of investigation if its role is in maintenance of the stem cell niche or is its primary role in controlling early neural development. Their work sheds light on Pauper's importance, but a more detailed understanding of its biological role will require further studies.

The in vitro work is quite solid. They have made a strong case for the formation of a regulatory complex that comprises Pax6-Pauper-KAP1. They show that Pauper is important in directing Pax6 and KAP1 to genomic locations. This is another example of a role for a long non-coding RNA as key component of a gene regulatory network that acts as a scaffold that aids in assembly of transcriptional machinery.

We thank the reviewer for the positive comments on the manuscript. We have performed many new experiments that we believe considerably enhance this study. In particular, these now show that *Pauper* is able to induce KAP1-dependent histone H3K9me3 changes at bound target genes *in trans*. We also discover a 4-fold enrichment in the overlap between *Pauper* and KAP1 shared binding sites genome-wide generating a more comprehensive view of the potential role of *Pauper* in mediating KAP1 genomic recruitment. In addition, we performed a more extensive analysis of the role of *Pauper* and KAP1 in neurogenesis *in vivo*. This, in particular, shows that both *Pauper* and KAP1 loss-of-function alters olfactory bulb neuron morphology and disrupts neurogenesis *in vivo*.

Specific comments

Thank you for these specific comments (below). We have taken care to rewrite this manuscript accordingly and have corrected the indicated typos.

1. The introduction spends too much effort on the general importance of lncRNAs and not enough on the background of Pauper (and Pax6).

2. Adhere to convention in Results to use past tense in describing results. See page 9, para 2 for examples, but this should be corrected throughout.

3. Also need to adhere to convention on gene names. Sometimes they use 'PAX6' and others it is 'Pax6'. All this work was done in mouse, so lower case and italics for gene names is appropriate. We followed the guidelines for mouse gene names and proteins as described in MGI for this study. http://www.informatics.jax.org/mgihome/nomen/gene.shtml#gene_sym Specifically "Protein symbols use all uppercase letters".

4. P6, para 2: "P4" should be spelled out as "passage 4"

P4 SVZ refers to postnatal day 4 in this context. We have spelt this out in the revised text. Sorry for the confusion.

5. P6, para 2: for clarity, change "N2A cells" to "mouse N2A neuroblastoma cells"

6. P6, para 3: The second sentence is quite vague here. But the topic is better handled later (p7, para 3). It needs revision or just wait till the subsequent section to describe that GFP is marking transfected cells, and explain why the majority of cells are radial glial-like.

In the neonatal SVZ the neural stem cells are radial glia. They gradually transform into SVZ astrocyte-like stem cells. To take this transition into account and any potential changes induced by the functional KD studies we decided to use the term "radial glia-like". We have rewritten the description of the electroporation technique to point out that "Neonatal SVZ stem cells are the cells lining the ventricles postnatally and thus are initially targeted by electroporation with TAPs appearing after 3 days and neuroblasts after one week (Boutin et al., 2008a, Chesler et al., 2008)".

7. P7, para1: no need to spell out "hpe" again

8. P8, para 3: "native RNA-IP experiments" is jargon. Needs to spell out for general readership.

9. P11 para 2: "CHART-seq mapped" is jargon and the topic is presented too vaguely here. Consider addressing the CHART-seq information in the introduction.

10. Methods: for clarity state that CD1 pups are mice.

Referee #3:

In this work, Sun and colleagues describe the identification of a novel ribonucleoprotein complex in neuroblastoma N2A cells comprising the lnc RNA Paupar, KAP1 and PAX6, the deposition of which at target genes is promoted by Paupar in trans, with impact on neural stem cell self-renewal and differentiation.

However, the demonstration suffers from many shortcomings and the proposed model is very premature. In particular, the Paupar-KAP1 connection needs major strengthening through more advanced chromatin and biochemical analyses, and the acquisition of data that address more directly KAP1 function in this complex, including through the use of previously characterized KAP1 mutants.

We thank the reviewer for his/her helpful comments. We have taken the intervening time to perform a considerable number of new experiments (described below) that, we believe, greatly strengthen this 'Paupar-KAP1 connection':

- In addition to our experiments analysing *Paupar* loss-of-function in the SVZ we have now depleted *Kap1* expression in the SVZ, as described in the revised version of the manuscript. This enabled us to investigate whether *Paupar* and *Kap1* can both regulate the same neurodevelopmental process in vivo. These experiments showed that loss-of-function of *Paupar* and of *KAP1* both led to similar changes in olfactory bulb neuron morphology thus providing further evidence that *Paupar* and *Kap1* functionally interact *in vivo*.
- We performed KAP1 ChIP-seq experiments and examined the overlap between KAP1 and *Paupar* chromatin binding which permitted a more comprehensive view of the potential role of *Paupar* in regulating KAP1 genomic recruitment. This analysis yielded a 4-fold enrichment of overlap between *Paupar*-KAP1 bound sequences (see below).

Other, specific comments:

1. Figure 1:

- Fig. 1ab: without controls (other transcripts, normalisation, other tissues, testing NS and SVZ in same experiment) these data are not informative. What is the error bar? Why is it absent in b? The impact on *Paupar* should be quantified, not just assumed (particularly relevant since invoking dose-dependance). Figure 1a, 1b are missing statistics and ticks on Y axis (present otherwise in Figures 3 and 4).

We apologise for our oversight and now present *Paupar* expression relative to the housekeeping gene *Gapdh* in a new figure (Supplemental Fig 3a). This also enabled us to compare *Paupar* relative expression in the SVZ, neurospheres and N2A cells. To more precisely quantify the impact on *Paupar* we electroporated neurospheres cultured from P4 SVZ with the *Paupar* targeting shRNA expression vectors and measured changes in *Paupar* expression three days later using RT-qPCR (Supplemental Fig 3b). This result confirmed the efficiency of these vectors in depleting *Paupar*

expression in agreement with our previous observations in N2A cells: specifically, sh165 caused robust *Paupar* knockdown whereas sh408 moderately reduced *Paupar* expression which we exploited to identify dose-dependent regulatory effects.

- Fig 1c is not useful.

Agreed: this has been removed from the manuscript.

- Fig. 1d: what does "electroporated GFP+ cells" mean? Nowhere is it specified the nature of the shRNA vectors, and whether they express GFP. Ki67 staining is invisible.

We apologise for this. We now have added text stating "All shRNA plasmids also express GFP" to the figure legend (now Fig 5). We also now provide a more informative Ki67 image.

- Fig. 1e: what does " $N \geq 3$ " mean? What techniques was used to determine that a GFP+ cell expressed or not the indicated markers?

We now indicate the number of experiments in the legend. The co-labelling is now described in the methods.

- Fig. 1f: $N = ?$

We now indicate the number of experiments.

- Fig. 1h: same as 1e. Where are "the small arrows"?

This is now explained in the figure legend.

- Was it impossible to verify the specificity of the *Paupar* KD via complementation?

We have not tried to do this in this study and instead used two shRNA targeting vectors to deplete *Paupar* expression.

2. Figure 2:

- 2c: RNA IP experiments are notoriously subjected to false positive. KAP1 mutant proteins could serve as useful controls.

We agree that native RNA-IP experiments, performed without cross-linking, are particularly and frequently subject to false positive findings. To test the specificity of the *Paupar*-KAP1 interaction we performed UV crosslinked RNA-IP (UV-RIP) experiments in addition to traditional native RNA-IPs (Figure 1). As the reviewer will know, UV-RIP uses UV to covalently cross-link short range RNA-protein associations and employs highly stringent washing of immobilised antibody-protein-RNA complexes to reduce both indirect associations and non-specific interactions formed in solution. Using UV-RIP we were able to validate the interaction between *Paupar* and KAP1 (Fig 1d). Furthermore, we observed this to be a direct interaction because the association of *Paupar* with KAP1 is reduced in the absence of UV treatment (Fig 1e).

- 2f: KAP1 alone is brought down by the flag antibody, indicating lack of specificity under the conditions used here. The PAX6 input lane should be shown. An IgG control should be used.

We reasoned that transfection of KAP1 expression vector alone serves as the most appropriate negative control for this FLAG IP experiment. This control identifies the amount of KAP1 that non-specifically binds to the FLAG beads. We tried increasing the salt concentration in the IP buffer as well as the salt and detergent in the wash buffer to reduce binding of KAP1 to beads alone yet were unable to reduce this background binding further. Despite this, Fig 1f clearly shows an enrichment of KAP1 binding in complex with transfected FLAG-PAX6 (lane 4) compared to KAP1 alone in the absence of transfected PAX6 (Lane 3). PAX6 is enriched on the beads following IP enabling straightforward detection. We were unable to detect FLAG-PAX6 in the input lane (at 1% input) so for clarity this was removed from the figure.

- Figure 2c, d, e missing statistics and ticks on Y axis;

Sorry for this. Statistics and y axis ticks have been added to all figures.

3. Figure 3:

- Complementation with WT or mutant KAP1 would be nice.

The effect of knockdown *Paupar* and *Kap1* together, as well as complementation, are planned for future studies.

- Without KAP1 ChIP-Seq data in these cells, the finding of up- and down-regulated genes upon KAP1 KD does not primarily suggest that KAP1 can function as either an activator or a repressor, but rather a superimposition of primary and secondary effects.

We agree that identification, via microarrays, of genes whose expression changes upon *Kap1* depletion will identify both direct and indirect targets. We do not know then know from this data whether KAP1 preferentially functions as an activator or repressor in complex with *Paupar* in proliferating N2A cells. Our new results in Fig 3 show that *Paupar*-KAP1-PAX6 complex is able to repress the transcriptional activity of bound regulatory elements *in trans*. This is because KAP1 recruitment correlates with H3K9me3 deposition at specific loci. This is consistent with our previous study showing that *Paupar* knockdown increased the activity of a bound sequence at the *E2f2* gene in a reporter assay. This is included in a modified Discussion. We do not know whether KAP1 can function as an activator at other loci in complex with *Paupar* but intend to investigate this specifically in future studies

- Why do the authors define the shRNA control as "scramble" in Figure 1 and "sh Cont" in Figure 3? Even though different scramble sequences might have been used, this is confusing. We fixed the ambiguous nomenclature in these figures. We apologise for the confusion.

- Figure 3b is missing statistics and Y axis ticks.

Sorry for this. Statistics and y axis ticks have been added to all figures.

- Figure 3c: A fold-change of 1.4 is really low; most investigators will use 2 as minimal threshold for differential gene expression.

We used a $\log_2 > 0.5$ fold change (1.4-fold) and a 5% FDR threshold to identify differentially expressed genes. These cut-offs are both in common use in gene expression studies.

4. Figure 4: Some positive controls should be used for the KAP1 ChIP-PCR (e.g. the 3' end of *KRAB-ZFP* genes, so as to get a better appreciation of the relative enrichment at tested loci.

In response, we compared enrichment of KAP1 binding at the *Mab21L2*, *Mst1*, *E2f2* and *Igfbp5* genes to binding at the *Zfp382* 3' UTR. This was identified as an exemplar KAP1 high affinity binding site at the 3' end of a ZFP gene using our KAP1 ChIP-seq experiments in N2A cells. Results showed that KAP1 binding to these regions is only 2- to 4-fold reduced compared to *Zfp382* 3' UTR.

5. Figure 5: This control becomes even more relevant here. Sites need to be explored, where KAP1 recruitment is *Paupar*-independent. The only one here is *Ezh2BS*, which is *Paupar*+ by *CHART*... A ChIP-Seq would be far more valuable than the presented ChIP-PCR, as it would give a global view of KAP1 recruitment in *Paupar*-high and *Paupar*-low conditions, and also performed following *Pax6* manipulations.

As suggested by the reviewer, we performed KAP1 ChIP-seq experiments to map KAP1 genome wide occupancy in N2A cells. We then examined the overlap between KAP1 and *Paupar* chromatin binding sites seeking to generate a more comprehensive view of the potential roles of *Paupar* in regulating KAP1 genomic recruitment. These experiments identified 46 KAP1 binding sites that are co-occupied by *Paupar*. This is a statistically significant 4-fold enrichment of overlap between *Paupar*-KAP1 bound sequences (GAT analysis). This finding indicates that *Paupar* specifically co-occupies a subset of KAP1-bound locations genome-wide and raises the possibility that KAP1 may interact with numerous additional lncRNAs to target specific locations in the genome. Future experiments will investigate this hypothesis.

- Figure 5a, an error bar (presumably *Scr*) is out of the chart.

To examine *Paupar* knockdown levels, results were normalised to the *Gapdh* reference gene and then presented relative to the *Scr* control in each experiment. *Paupar* expression in *Scr* was set arbitrarily at 1 in each case so does not have an error bar. We are interested in the relative change in *Paupar* expression.

- Figure 5d and 2f are in clear conflict. In figure 5d following FLAG-PAX6 transfection the authors detect KAP1 in pull-down complexes but the same was not observed in Figure 2f (top panel, second lane). Why do they overexpress exogenous KAP1 whereas this protein is already present at good levels in N2A cells?

We apologise for the confusion here. Both these experiments (Fig 1f and 3f in the new manuscript) use transfected KAP1 protein to investigate the interaction with FLAG-PAX6. We have modified Fig 1f to make this clearer. We now state in the figure legend that cells are transfected with FLAG-PAX6 and KAP1 expression vectors.

We used transfected KAP1 protein because under the conditions used in the IP experiment we were unable to detect an interaction between endogenous PAX6 and KAP1 in solution. We believe that this is because the interaction occurs on chromatin. This hypothesis is consistent with results of our KAP1 ChIP experiments showing that KAP1 can associate with PAX6-bound sequences on chromatin (Fig 3B).

Analysis of the intersection of KAP1, PAX6 and *Paupar* regulated genes provides further evidence for a functional KAP1, PAX6 and *Paupar* regulatory complex. This identified 87 genes that are regulated by these 3 factors, a number that is 35.8-fold greater than expected by random sampling.

6. Discussion:

- The authors cannot claim that KAP1 interacts directly with *Paupar*, since they did not perform binding assays with purified recombinant protein and RNA.

Findings from the UV-RIP experiments (Figure 1) in the revised manuscript provide strong evidence that *Paupar*-KAP1 directly interact in N2A cells in culture. See response to point 2.

- More generally, the authors say very little about the functional role of KAP1, in a putative *Paupar*-*Pax6*-KAP1 complex. If they think it acts as a coactivator owing to some post-translational modification, they should test this (probing for pS473 and pS824 KAP1 by ChIP, co-IP, etc.).

- The recruitment of KAP1 to the 3' end of zinc finger genes is KRAB-mediated (a statement on page 16 suggests otherwise).

In the revised manuscript we show that *Paupar* is able to promote repressive H3K9me3 changes at selected bound sequences in *trans*. This suggests that KAP1 is acting as a corepressor at the genes tested. However, we do not know whether KAP1 can function as an activator at all other loci when in complex with *Paupar*. This is a subject for further exploration in the future. We have extensively modified the Discussion in the new manuscript and in-so-doing have removed the text describing KAP1 phosphorylation in order that we focus our discussion specifically on the data described in the paper.

2nd Editorial Decision

21st November 2017

Thank you for submitting a new version of your manuscript for consideration by the EMBO Journal and my apologies for the extended duration of the review process. Your study has now been seen by two of the original referees and their comments are shown below.

As you will see from the reports, both referees express interest in the findings reported in your manuscript and acknowledge the improvements you have made from the previous version. However, you will see that they also point to several issues that haven't been adequately addressed, meaning that the study requires additional revision before the referees can support publication of the manuscript in The EMBO Journal.

More specifically, you will see that ref #1 (old referee #3) finds that the overall presentation and analysis of the data is much improved but that it remains unclear how many target genes show binding by all three factors in the complex (ie the generality of the regulatory mechanism described here)

Ref #2 (old referee #1) raises more serious concerns about the revised manuscript and finds that the

phenotypic analysis upon Paupar depletion in vivo remains inconclusive. This calls into question how and to what extent Paupar (and the complex with Pax6 and KAP1) control neurogenesis in vivo, which could in our view undermine a substantial section of the study.

Since the referees thus question both the generality (ref #1) and in vivo relevance (ref #2) of Paupar-KAP1-Pax6 axis I conducted an additional round of consultation with them based on the reports (and asking about these specific points). This resulted in the following feedback from the refs:

Ref #1 was not too concerned about the absolute number of targets since this to some extent depends on the threshold used for the bioinformatic analysis but restated that the main concern is the undefined cellular effects on neurogenesis. Upon further prompts from my side on whether the data in fig 5 and 6 in its current form still supports a functional link between Paupar and KAP1 or if it should rather be removed from the study altogether, the referee responded:

'I find it reasonable that, yes, some effect may be there but which effect this is remains totally unclear. Perhaps the best is to request the authors to tone this down and specifically write that what they see might be due to many factors that were not investigated (e.g migration and so on)'

Ref #2 elaborated on the comment about shared binding sites with the following statement:

'Regarding the KAP1-Paupar-Pax6 axis, the evidence is still a weak point of the paper. For one, I do not think that the UV-sensitivity of the co-immunoprecipitation demonstrates a direct interaction. Also, the overlap between the KAP1 and Paupar genomic recruitments, with so many sites for both, is not overwhelming. It would have been useful to document sites where all three factors are bound (eg with screen shots to see what the "overlap" is and whether it is consistent with KAP1 docked via Paupar), with nearby genes deregulated when one or the other is missing, as well as sites where KAP1 recruitment is abrogated when Paupar is downregulated.'

As you may know, it is generally EMBO Journal policy to allow only a single round of revision, but given the referees' overall interest in the study and their positive recommendations I'm willing to make an exception from that rule and let you have an additional round of major revision in this case. However, this also means that it'll be essential for you to address the remaining concerns (clarifying the extent of genome-wide complex binding and toning down the conclusions for neurogenesis) before we can take any final steps towards publication.

In addition, I want to add another technical point that wasn't directly brought up by the referees but which I think it would nonetheless be helpful to discuss in the manuscript. Your study deals with a supposedly nuclear lncRNA but all depletion experiments are done using shRNAs. I realise that the efficiency of these in the nucleus (if any) is a debated issue in the field (where there is not necessarily a clear consensus) but it would in my view be helpful to support this with antisense oligos that have a better activity in targeting nuclear RNAs. Alternatively, please discuss/comment on this issue in the revised manuscript.

REFEREE REPORTS

Referee #1:

The authors extensively modified the paper, adding several experiments. Moreover, the presentation now looks much more rigorous, with proper labelling on the graphs and statistical analyses of the data. As a result of their effort, the story is now quite convincing, although assessing its in vivo aspects is beyond our expertise.

Still one point to be made: Line 208, what do the authors mean by "not enriched for KRAB-ZNF transcription factor binding motifs"? There are hundreds of KRAB-ZNFs in the genome, they do not have a single binding motif.

The authors determine that KAP1 and Paupar control the same targets only when forming a ternary complex with Pax6. They then go on and define the genome wide occupancy of KAP1 and Paupar but they do not include Pax6 in this analysis. As these would probably be the functional targets of the ternary complex it would be interesting to know the genomic regions where all three colocalize. Does the localization correlate with the functional overlap of KAP1/Pax6/Paupar in Figure 3a?

Referee #2:

Concerning the revised manuscript by Pavlaki et al., i confirm my general opinion about the novelty and interest in this study. The authors have gone through a substantial revision adding new and important data addressing most, though not all, my concerns regarding their earlier version.

Having said this, and appreciating the authors' succesful efforts to improve their work, i still have to confirm my reservation with regard to understanding the phenotype reported on neurogenesis. Specifically in relation to my previous point 2 and its i-vi sub-points, i cannot fail to notice that most were either not, or only partially, addressed (i, iii, v) while others were overcome by, simply, removing the original data (i, vi) and again others have limited validity (i.e.: iv that seems not to apply in the case of quantifications in Fig 6). Again despite the additional data and improvements offered, in this revision I still fail to understand the meaning and reason for the differences in "neurogenic/stem cell" phenotypes observed. As one example, Fig 5 addresses this but my earlier opinion about shortcomings still remain overall valid including the quality and resolution of the immunohistochemistry shown, slight inconsistencies in various parameters measured by the 2 shRNAs overtime and lack of data concerning key parameters, such as neuronal migration, quiescence and others the remain unaddressed (... I understand the reasoning for not doing these experiments, as explained in the authors' rebuttal, but the fact remains that the phenotype(s) are not compellingly dissected).

With all that, I consider this a solid and interesting work particularly valid with regard to the molecular aspects about the function of this lncRNA but whose significance for neurogenesis and cell biological aspects of stem cell differentiation and brain development are circumstantial, indirect, or poor.

The editors must decide in this case which of these pros and cons would need to be prioritized in the frame of the purpose of this journal.

Thank you for submitting a new version of your manuscript for consideration by the EMBO Journal and my apologies for the extended duration of the review process. Your study has now been seen by two of the original referees and their comments are shown below.

As you will see from the reports, both referees express interest in the findings reported in your manuscript and acknowledge the improvements you have made from the previous version. However, you will see that they also point to several issues that haven't been adequately addressed, meaning that the study requires additional revision before the referees can support publication of the manuscript in The EMBO Journal.

More specifically, you will see that ref #1 (old referee #3) finds that the overall presentation and analysis of the data is much improved but that it remains unclear how many target genes show binding by all three factors in the complex (ie the generality of the regulatory mechanism described here).

Ref #2 (old referee #1) raises more serious concerns about the revised manuscript and finds that the phenotypic analysis upon Paupar depletion in vivo remains inconclusive. This calls into question how and to what extent Paupar (and the complex with Pax6 and KAP1) control neurogenesis in vivo, which could in our view undermine a substantial section of the study.

Since the referees thus question both the generality (ref #1) and in vivo relevance (ref #2) of Paupar-KAP1-Pax6 axis I conducted an additional round of consultation with them based on the reports (and asking about these specific points). This resulted in the following feedback from the refs:

Ref #1 was not too concerned about the absolute number of targets since this to some extent depends on the threshold used for the bioinformatic analysis but restated that the main concern is the undefined cellular effects on neurogenesis. Upon further prompts from my side on whether the data in fig 5 and 6 in its current form still supports a functional link between Paupar and KAP1 or if it should rather be removed from the study altogether, the referee responded: 'I find it reasonable that, yes, some effect may be there but which effect this is remains totally unclear. Perhaps the best is to request the authors to tone this down and specifically write that what they see might be due to many factors that were not investigated (e.g migration and so on)'

Ref #2 elaborated on the comment about shared binding sites with the following statement: 'Regarding the KAP1-Paupar-Pax6 axis, the evidence is still a weak point of the paper. For one, I do not think that the UV-sensitivity of the co-immunoprecipitation demonstrates a direct interaction. Also, the overlap between the KAP1 and Paupar genomic recruitments, with so many sites for both, is not overwhelming. It would have been useful to document sites where all three factors are bound (eg with screen shots to see what the "overlap" is and whether it is consistent with KAP1 docked via Paupar), with nearby genes deregulated when one or the other is missing, as well as sites where KAP1 recruitment is abrogated when Paupar is downregulated.'

As you may know, it is generally EMBO Journal policy to allow only a single round of revision, but given the referees' overall interest in the study and their positive recommendations I'm willing to make an exception from that rule and let you have an additional round of major revision in this case. However, this also means that it'll be essential for you to address the remaining concerns (clarifying the extent of genome-wide complex binding and toning down the conclusions for neurogenesis) before we can take any final steps towards publication.

Thank you very much for the detailed review of our manuscript and succinct suggestions to address the remaining issues. In summary, we have performed the following analyses which we believe have further improved the study:

(1) We carried out PAX6 ChIP-qPCR to measure PAX6 occupancy at a subset of the ChIP-seq and CHART-seq defined KAP1-Paupar bound sequences and further assess the generality of the Paupar-KAP1-PAX6 complex as requested. The results (see point by point response for details)

provide strong evidence that the majority of *Paupar*-KAP1 co-occupied binding sites are also bound by PAX6.

(2) We have toned down the conclusions for neurogenesis as suggested. In the original paper we analysed the effect of *Paupar* and KAP1 knockdown both in the SVZ (old Fig 5), where the adult neural stem cells reside, and also in the olfactory bulb (old Fig 6), where the differentiated neurons that have migrated from the SVZ are located. We agree that we do not fully understand the function of *Paupar* in the SVZ so have removed this section. The revised manuscript now describes the robust phenotype we observed in the OB in order to highlight the key point that *Paupar* knockdown disrupts neurogenesis in vivo.

Given these changes the new manuscript therefore has a greater emphasis on the molecular description of *Paupar* action.

In addition, I want to add another technical point that wasn't directly brought up by the referees but which I think it would nonetheless be helpful to discuss in the manuscript. Your study deals with a supposedly nuclear lncRNA but all depletion experiments are done using shRNAs. I realise that the efficiency of these in the nucleus (if any) is a debated issue in the field (where there is not necessarily a clear consensus) but it would in my view be helpful to support this with antisense oligos that have a better activity in targeting nuclear RNAs. Alternatively, please discuss/comment on this issue in the revised manuscript.

We agree that the efficiency of using shRNAs to deplete the expression of nuclear lncRNAs is debated. To address this, we previously performed fractionation experiments and quantified *Paupar* transcript levels in different subcellular fractions following shRNA transfection. This confirmed that *Paupar* expression in the nucleus, including in the chromatin associated fraction, is reduced. The data is included in Supplementary Figure S2 of our previous EMBO J paper in 2014 where we first described *Paupar*. We note that we have also reduced expression of other nuclear lncRNAs using shRNAs. See *Dali* example in Chalei et al., 2014, eLife.

When preparing your letter of response to the referees' comments, please bear in mind that this will form part of the Review Process File, and will therefore be available online to the community. For more details on our Transparent Editorial Process, please visit our website: http://emboj.embopress.org/about#Transparent_Process

Thank you for the opportunity to consider your work for publication. I look forward to your revision.

Referee #1:

The authors extensively modified the paper, adding several experiments. Moreover, the presentation now looks much more rigorous, with proper labelling on the graphs and statistical analyses of the data. As a result of their effort, the story is now quite convincing, although assessing its in vivo aspects is beyond our expertise.

We thank the reviewer for recognising the extensive modifications that we made to improve the original submission

Still one point to be made: Line 208, what do the authors mean by "not enriched for KRAB-ZNF transcription factor binding motifs"? There are hundreds of KRAB-ZNFs in the genome, they do not have a single binding motif.

We previously used *de novo* motif discovery to search for sequences that are enriched in *Paupar* bound locations genome-wide. This analysis did not identify enrichment of a sequence that resembles any KRAB-ZNF binding motif. We therefore modified this sentence to read “*Paupar* bound sequences are preferentially located at gene promoters and are not enriched for KRAB-ZNF transcription factor binding motifs as determined using *de novo* motif discovery”. Our previous publication showing this is referenced.

The authors determine that KAP1 and Paupar control the same targets only when forming a ternary complex with Pax6. They then go on and define the genome wide occupancy of KAP1 and Paupar but they do not include Pax6 in this analysis. As these would probably be the functional targets of the ternary complex it would be interesting to know the genomic regions where all three colocalize. Does the localization correlate with the functional overlap of KAP1/Pax6/Paupar in Figure 3a?

To address this, we performed PAX6 ChIP-qPCR to further assess the generality of the *Paupar*-KAP1-PAX6 complex. We measured PAX6 occupancy at a subset of 15 out of the 46 of ChIP-seq and CHART-seq defined KAP1-*Paupar* bound sequences. The results identified statistically significant PAX6 enrichment at 11 out of 15 (73%) locations tested compared to an IgG control (see Fig 4d in the revised manuscript). Furthermore, we discovered that 4 of these 11 binding sites lie within putative GREAT defined regulatory regions of genes (*Tshz2*, *Syt7*, *Syt1* and *Npy*) whose expression changes when PAX6 expression is depleted in N2A cells using microarray data and are thus likely to be direct PAX6 targets. These results therefore demonstrate that PAX6 associates with the majority (11 out of 15) of the KAP1-*Paupar* co-occupied sequences that we tested and suggest that PAX6 is likely to play a regulatory role at a significant number of KAP1-*Paupar* co-occupied sequences genome-wide.

Referee #2:

Concerning the revised manuscript by Pavlaki et al., i confirm my general opinion about the novelty and interest in this study. The authors have gone through a substantial revision adding new and important data addressing most, though not all, my concerns regarding their earlier version.

Having said this, and appreciating the authors' successful efforts to improve their work, i still have to confirm my reservation with regard to understanding the phenotype reported on neurogenesis. Specifically in relation to my previous point 2 and its i-vi sub-points, i cannot fail to notice that most were either not, or only partially, addressed (i, iii, v) while others were overcome by, simply, removing the original data (i, vi) and again others have limited validity (i.e.: iv that seems not to apply in the case of quantifications in Fig 6). Again despite the additional data and improvements offered, in this revision I still fail to understand the meaning and reason for the differences in "neurogenic/stem cell" phenotypes observed. As one example, Fig 5 addresses this but my earlier opinion about shortcomings still remain overall valid including the quality and resolution of the immunohistochemistry shown, slight inconsistencies in various parameters measured by the 2 shRNAs overtime and lack of data concerning key parameters, such as neuronal migration, quiescence and others the remain unaddressed (... I understand the reasoning for not doing these experiments, as explained in the authors' rebuttal, but the fact remains that the phenotype(s) are not compellingly dissected).

With all that, I consider this a solid and interesting work particularly valid with regard to the molecular aspects about the function of this lncRNA but whose significance for neurogenesis and cell biological aspects of stem cell differentiation and brain development are circumstantial, indirect, or poor.

We appreciate these comments and have toned down the conclusions for neurogenesis (see below) to focus more on the molecular mechanism of *Paupar* action in the revised manuscript.

The editors must decide in this case which of these pros and cons would need to be prioritized in the frame of the purpose of this journal.

We agree with this reviewer that we do not fully understand the function of *Paupar* in the SVZ. In particular, we recognise that the exact effect of *Paupar* knockdown on stem cell maintenance and/or neuronal differentiation and migration still remains unclear. We have therefore removed the SVZ data from the revised manuscript and instead describe the clear phenotype that we observe in the olfactory bulb upon both *Paupar* and KAP1 knockdown (Fig 5 in the revised manuscript). This shows that both *Paupar* and KAP1 depletion lead to a reduction in the number of differentiated neurons that have migrated there and that the olfactory bulb neurons in the knockdown cells display altered morphology. The data also shows that the decrease in neuronal numbers is not due to increased apoptosis caused by *Paupar* or KAP1 knockdown and we discuss that the reduction in numbers could be caused by a decrease in migration and/or delay in differentiation in a revised discussion section. By toning down our analysis of the function *Paupar* in neurogenesis along these lines, we believe that the data more convincingly illustrates the key finding that *Paupar* and KAP1 are needed for proper olfactory bulb neurogenesis.

3rd Editorial Decision

22nd February 2018

Thank you for submitting a revised version of your manuscript; it has now been seen by one of the original referees whose comments are shown below. As you will see the referee appreciates the new data you have included on Pax6 binding and suggests a few additional analysis points that would integrate that data better with the rest of the study.

I realise that some of the comments amount to new points that were not raised in earlier rounds of review and I will therefore not insist on their completion for acceptance of your manuscript in The EMBO Journal. As such, you do not have to address points I and IV, since they would involve the generation of new experimental data. However, if it is not too time-consuming at this point - and since the data is there already - I would suggest that you include the classification of positively and negatively regulated gene suggested by the referee in points II and III. Feel free to contact me about any questions about this.

Based on the overall positive feedback from the referee I would like to invite you to submit a final version of the manuscript in which you include the minor points outlined above.

REFeree REPORTS

Referee #1:

In the manuscript " Paupar LncRNA promotes KAP1 dependent chromatin changes and regulates olfactory bulb neurogenesis ", K. Vance and colleagues show an interesting new mode of recruitment for KAP1 at gene promoters, mediated by the LncRNA Paupar. This is interesting and novel, as the mode by which KAP1 is recruited at promoter in an RBCC-independent manner remained unknown.

Although the role of Paupar in the recruitment of KAP1 at the promoters of several genes is well supported by the data presented, the authors should present in a systematic way what are the functional implication of this complex at bound loci. We highlight the major suggestions in roman numbers. In Figure 2, the authors show that Paupar and KAP1 share several target genes and most of these are regulated in a positive fashion by the two factors. The authors then proceed to demonstrate that Paupar promotes KAP1 binding to chromatin on loci that are also bound by PAX6. They then show that Paupar-KAP1-PAX6 induce deposition of H3K9me3 at bound loci. (I) The authors should show how expression of this targets is affected by removal of any of the elements of the ternary complex. The authors claim that the majority of the Paupar-KAP1 bound loci are co-regulated by PAX6, with this being a repressive complex (brings heterochromatin). (II) The authors should elaborate on why the Paupar and KAP1 positively regulate the majority of their shared target (Figure 2E) but the Paupar-KAP1-PAX6 targets are negatively regulated by the complex. I believe the data are there, for example the authors could (III) compare the subsets of genes in Figure 2E with the intersection in Figure 3A or Figure 4B. This would allow a classification of Paupar-KAP1 targets, positively regulated, and Paupar-KAP1-PAX6 targets, negatively regulated (which should be shown in point I). The author could also (IV) take as example *Ezh2* which is targeted by Paupar-KAP1, not by PAX6 and not enriched in H3K9me3. I believe the paper would benefit much from a more

organized representation of the data, to understand how the different subsets of Paupar/KAP1/PAX6 regulated targets are controlled.

Minor points:

- " Figure 3C and Figure 3G need statistical analyses.
- " Figure EV4 A and C need proper labelling.

3rd Revision - authors' response

5th March 2018

Referee #1:

In the manuscript "Paupar LncRNA promotes KAP1 dependent chromatin changes and regulates ilfactory bulb neurogenesis", K. Vance and colleagues show an interesting new mode of recruitment for KAP1 at gene promoters, mediated by the LncRNA Paupar. This is interesting and novel, as the mode by which KAP1 is recruited at promoter in an RBCC-independent manner remained unknown.

Thank you for your interest in this study and the positive assessment.

Although the role of Paupar in the recruitment of KAP1 at the promoters of several genes is well supported by the data presented, the authors should present in a systematic way what are the functional implication of this complex at bound loci. We highlight the major suggestions in roman numbers. In Figure 2, the authors show that Paupar and KAP1 share several target genes and most of these are regulated in a positive fashion by the two factors. The authors then proceed to demonstrate that Paupar promotes KAP1 binding to chromatin on loci that are also bound by PAX6. They then show that Paupar-KAP1-PAX6 induce deposition of H3K9me3 at bound loci. (I) The authors should show how expression of this targets is affected by removal of any of the elements of the ternary complex. The authors claim that the majority of the Paupar-KAP1 bound loci are co-regulated by PAX6, with this being a repressive complex (brings heterochromatin). (II) The authors should elaborate on why the Paupar and KAP1 positively regulate the majority of their shared target (Figure 2E) but the Paupar-KAP1-PAX6 targets are negatively regulated by the complex. I believe the data are there, for example the authors could (III) compare the subsets of genes in Figure 2E with the intersection in Figure 3A or Figure 4B. This would allow a classification of Paupar-KAP1 targets, positively regulated, and Paupar-KAP1-PAX6 targets, negatively regulated (which should be shown in point I). The author could also (IV) take as example Ezh2 which is targeted by Paupar-KAP1, not by PAX6 and not enriched in H3K9me3. I believe the paper would benefit much from a more organized representation of the data, to understand how the different subsets of Paupar/KAP1/PAX6 regulated targets are controlled.

As requested, we examined the intersection between the 46 Paupar-KAP1 co-bound locations and the 244 Paupar-KAP1 co-regulated genes to categorise different subsets of direct targets. This only identified shared binding sites within the regulatory regions of four genes and therefore we were unable to perform a large scale classification. It should be noted though that we expect this to be an under-representation of the total number of direct Paupar-KAP1 co-regulated targets given the complex cause-and-effect relationship between histone H3 methylation and gene expression (see below). Text changes addressing this are found on Page 12: Lines 274-279 of the new manuscript. It is important to note that the proposed analysis to identify different classes of positively and negatively regulated genes was suggested based on the assumption that "Paupar-KAP1-PAX6 targets are negatively regulated by the complex". However, the results show that this is not the case and we clarify our findings in the revised submission.

Fig 3 indicates that Paupar depletion leads to a reduction in KAP1 chromatin association and H3K9me3 deposition at three Paupar-KAP1-PAX6 shared target genes (*Mab21L2*, *Mst1*, *E2f2*). Using our previously published microarray dataset of Paupar targets we now report that the expression of two of these genes (*Mab21L2*, *Mst1*) is down-regulated upon Paupar depletion suggesting that they are activated by the Paupar-KAP1-PAX6 complex whilst the other gene (*E2f2*) is up-regulated, suggesting that it is repressed by this complex (see Page 11: Lines 251-258). H3K9me3 deposition at these genes therefore does correlate with repression. Although surprising, these findings are consistent with recent work using dCas9 fusion proteins to target histone

methylation to specific loci (O'Geen *et al*, 2017) and suggest a complex relationship between *Paupar* mediated KAP1 dependent chromatin changes and gene expression. We do not fully understand the mechanistic basis of this and plan to investigate this further in the future.

Minor points:

"Figure 3C and Figure 3G need statistical analyses.
Done.

"Figure EV4 A and C need proper labelling.
These have been fixed. Thanks.

Corresponding Author Name: Keith Vance

Journal Submitted to: EMBO J

Manuscript Number: EMBOJ-2017-98219